# Reticuline and Coclaurine Exhibit Vitamin D Receptor-Dependent Anticancer and Pro-Apoptotic Activities in the Colorectal Cancer Cell Line HCT116

**DOI:** 10.3390/cimb47100810

**Published:** 2025-10-01

**Authors:** Hind A. Alghamdi, Sahar S. Alghamdi, Maryam Hassan Al-Zahrani, Thadeo Trivilegio, Sara Bahattab, Rehab AlRoshody, Yazeid Alhaidan, Rana A. Alghamdi, Sabine Matou-Nasri

**Affiliations:** 1Biochemistry Department, Faculty of Science, King Abdulaziz University, Jeddah 21589, Saudi Arabia; haalghamdi@pnu.edu.sa (H.A.A.); mhsalzahrani@kau.edu.sa (M.H.A.-Z.); 2Blood and Cancer Research Department, King Abdullah International Medical Research Center, King Saud bin Abdulaziz University for Health Sciences (KSAU-HS), Ministry of National Guard-Health Affairs (MNG-HA), Riyadh 11481, Saudi Arabia; s.bahattab4@gmail.com (S.B.); alroshodyre@mngha.med.sa (R.A.); 3Chemistry Department, Princess Nourah bint Abdulrahman University, Riyadh 11671, Saudi Arabia; 4Pharmaceutical Sciences Department, College of Pharmacy, King Saud bin Abdulaziz University for Health Sciences (KSAU-HS), Ministry of National Guard-Health Affairs (MNG-HA), Riyadh 11481, Saudi Arabia; ghamdisa@ksau-hs.edu.sa; 5Medical Research Core Facility and Platforms, King Abdullah International Medical Research Center, King Saud bin Abdulaziz University for Health Sciences (KSAU-HS), Ministry of National Guard-Health Affairs (MNG-HA), Riyadh 11481, Saudi Arabia; thad3vleeu@gmail.com; 6Medical Genomics Research Department, King Abdullah International Medical Research Center, King Saud bin Abdulaziz University for Health Sciences (KSAU-HS), Ministry of National Guard-Health Affairs (MNG-HA), Riyadh 11481, Saudi Arabia; alhaidanya@mngha.med.sa; 7Department of Chemistry, Science and Arts College, King Abdulaziz University, Rabigh 25732, Saudi Arabia; raalghamdi3@kau.edu.sa; 8Regenerative Medicine Unit, King Fahd Medical Research Centre, King Abdulaziz University, Jeddah 21589, Saudi Arabia; 9Biosciences Department, Faculty of the School of Systems Biology, George Mason University, Manassas, VA 20110, USA

**Keywords:** colorectal cancer, vitamin D receptor, vitamin D_3_, coclaurine, reticuline, anticancer, apoptosis, molecular docking

## Abstract

Alkaloids have garnered significant interest as potential anticancer agents. Vitamin D receptor (VDR) plays a role in preventing the progression of colorectal cancer (CRC) and may be a crucial mediator of the anticancer effects produced by certain alkaloids. The search for novel anticancer drugs that induce VDR expression and act through the VDR could improve the clinical outcomes of CRC patients. The anticancer and pro-apoptotic effects of coclaurine and reticuline were investigated using CRISPR/Cas9-edited VDR/knockout (KO) and wild-type (WT) CRC HCT116 cell lines. Western blotting, RT-qPCR, confocal microscopy, cell viability, scratch assays, and flow cytometry were employed to assess VDR expression and cellular localization, cell growth, wound-healing, cytotoxicity, apoptotic status, cell cycle progression, and VDR-mediated gene expression. Coclaurine and reticuline dose-dependently inhibited HCT116-WT cell viability, decreased wound-healing, and increased VDR nuclear localization and gene expression while downregulating the oncogenic genes *SNAIL1* and *SNAIL2*. Both alkaloids induced late apoptosis in HCT116-WT cells, increased the cleavage of PARP and caspase-3, and upregulated Bax and *TP53* while decreasing *BCL-2*. Both alkaloids caused HCT116-WT cell growth arrest in the S-phase, which is associated with cyclin A1 overexpression. Coclaurine and reticuline lost their anticancer effects in HCT116-VDR/KO cells. Docking studies revealed that both alkaloids occupied the VDR’s active site. These findings demonstrate that coclaurine and reticuline exert anti-CRC and pro-apoptotic activities via the VDR, suggesting them as natural therapeutic candidates. The use of in vivo CRC models is needed to validate the anticancer activities of coclaurine and reticuline.

## 1. Introduction

Colorectal cancer (CRC) is the third most common primary and aggressive malignant tumor in humans, especially in men, and it is the fifth most deadly cancer in the world [1]. According to the World Health Organization (WHO), by 2040, the number of new CRC cases could exceed 3 million, and there could be more than 1.5 million related deaths worldwide [2]. The main CRC management modalities include surgery in resectable cases, radiotherapy, immunotherapy, and chemotherapy [3,4]. When detected late at an advanced stage, CRC is one of the most challenging cancer types to treat [5]. Low vitamin D receptor (VDR) expression is considered an adverse prognostic biomarker for CRC patients, making VDR expression a valuable biomarker for the diagnosis and prediction of CRC progression [6,7].

Vitamin D (VitD), particularly its active metabolite VitD_3_ (i.e., cholecalciferol), which is named 25-dihydroxycholecalciferol (i.e., 25(OH)D or calcitriol), exhibits anticancer properties by binding to its cytoplasmic receptor, VitD receptor (VDR), which translocates into the nucleus and acts as a transcription factor after forming a complex with retinoid X receptor and joining the VitD-response element of the targeted gene promoter region [8]. This ligand-activated transcription factor regulates the expression of genes responsible for cell growth, differentiation, epithelial–mesenchymal transition (i.e., *snail family transcriptional repressor* (*SNAIL1* and *SNAIL2*), metabolism (i.e., *silent mating-type information regulation 2 homolog* (*SIRT1*), apoptosis and signaling pathways (i.e., *phosphatidylinositol 3-kinase* (*PI3K*)/*protein kinase B* (*Akt*), *hypoxia-inducible factor* (*HIF*)*-1*, *forkhead box O* (*FoxO*)) [9,10]. Numerous epidemiological studies have reported that VitD deficiency (<20 ng/mL) is associated with a higher risk of CRC development and progression, while sufficient (30–35 ng/mL to 35–40 ng/mL) and high doses (>40 ng/mL) of serum levels of the VitD circulating form 25-(OH)D prevent CRC [11,12]. In a large cohort, CRC patients with high VDR expression levels had longer overall survival, presenting the VDR as a valuable prognostic marker, conversely, low VDR expression was associated with CRC progression and poor prognosis [6]. Thus, manipulations of VDR expression through genetic strategies, natural products, or drug-based strategies have been under investigation for the prevention of various diseases, including CRC [13,14].

*Annona muricata* (*A. muricata*) is an evergreen plant that is widely distributed in tropical and subtropical regions. Numerous phytochemicals, particularly alkaloids, extracted from *A. muricata* have garnered particular attention for their anti-inflammatory and anticancer properties against the liver, lung, prostate, pancreas, breast and colon cancers [15,16,17]. *A. muricata* produces seven isoquinoline alkaloids, including reticuline and coclaurine, primarily in its leaves, roots, and stem barks [18]. Reticuline is the important branch point in the biosynthesis of most benzylisoquinoline alkaloids [19]. Coclaurine has demonstrated anticancer activities against human CRC (HCT116) and breast cancer (MCF-7) cell lines in vitro [20]. A molecular docking study revealed molecular interactions between coclaurine, reticuline, and CRC receptors (i.e., TRAF2 and Nick interacting kinase (TNIK), vascular endothelial growth factor receptor (VEGFR), epidermal growth factor receptor (EGFR)), suggesting that both alkaloids could be potential anti-CRC drugs [21]. Moreover, berberine, a reticuline-derived alkaloid, enhances mucosal barrier function in newborn rats by promoting VDR activities [22], indicating the potential of some alkaloids in modulating VDR activities, including anti-CRC [13].

The discovery of novel drugs, including natural products, that upregulate *VDR* gene expression and act through the VDR would pave the way for the development of targeted therapeutic strategies for the management of CRC patients [13,23]. In this study, we aimed to investigate the anti-CRC properties of the alkaloids coclaurine and reticuline through the VDR by assessing the wound-healing process, cell viability, VDR cellular localization, and the gene expression of *VDR* and its target (i.e., *SNAIL1* and *SNAIL2*) in CRC cells using clustered regularly interspaced palindromic repeats (CRISPR)/Cas9-edited HCT116-VDR/knockout (KO) and wild-type (WT) HCT116 cell lines in comparison with the main VDR ligand, VitD_3_. In addition, we explored the CRC cell death mechanisms induced by coclaurine and reticuline. Here, the potential effects of coclaurine and reticuline on apoptosis and cell cycle analysis and their impact on the expression of genes/proteins related to apoptosis and the cell cycle were investigated using wild-type (WT) and VDR/knockout (KO) HCT116 cell lines. A prediction of the molecular interactions of the two alkaloids with the VDR’s active site was studied using an in silico approach.

## 2. Materials and Methods

### 2.1. Reagents

Coclaurine (#SC5960) and reticuline (#SR8320) with a purity ≥ 98% were extracted from *A. muricata* roots and provided by Solarbio Science and Technology Co., Ltd., Beijing, China). Mouse primary anti-VDR (#sc-13133), cyclin A1 (#sc-271645), cyclin B1 (#sc-70898), cyclin D1 (#sc-8396) antibodies were provided by Santa Cruz Biotechnology Inc., (Dallas, TX, USA). Rabbit primary anti-cleaved caspase-3 (#9664), pro-caspase 3 (#9665), cleaved poly (ADP-ribose) polymerase (PARP) (#5625), PARP (#9542), B-cell lymphoma 2 (Bcl-2, #15071S), BCL2-associated X (Bax, #2772S) monoclonal antibodies were purchased from Cell Signaling Technology (Danvers, MA, USA). Mouse anti-glyceraldehyde 3-phosphate dehydrogenase (GAPDH) monoclonal antibody (#AM4300) was from Invitrogen. All other reagents were provided by Thermo Fisher Scientific (Waltham, MA, USA) unless otherwise indicated.

### 2.2. Establishment of the CRC-VDR/Knockout Cell Line (HCT116-VDR/KO)

Human CRC expressing VDR wild-type (HCT116-WT) and VDR/KO (HCT116-VDR/KO) cell lines were provided by the American Type Culture Collection (Manassas, VA, USA) via Synthego Corporation (Menlo Park, CA, USA). The design and synthesis of the modified single guide RNA, predicted with minimal off-targets, and donor template (VDR/KO) were performed by Synthego Corporation CRISPR Genome Engineering Service. Briefly, the guide RNA Sequence was 5′-AUUCACCUGCCCCUUCAACG-3′. The guide RNA cut location was: chr12:47,865,137. VDR-specific guide RNAs were mixed with Cas9 to form a ribonucleoprotein (RNP, Appendix A). RNPs and the donor template were delivered into the HCT116 cell line through the electroporation setting using 200-point optimization (Synthego Corporation, Menlo Park, CA, USA), which enabled the establishment of the HCT116-VDR/KO cell line. In comparison with the HCT116-WT genome, CRISPR/Cas9-edited HCT116-VDR/KO was confirmed through Sanger sequencing (Appendix A). Moreover, 99% of the edited cells carried the desired mutation, with a 99% insertion/deletion (INDEL) frequency in the VDR gene according to an evaluation of CRISPR editing efficiency (Appendix A).

### 2.3. Cell Culture and Treatment

HCT116-WT and HCT116-VDR/KO cells were cultured in Roswell Park Memorial Institute (RPMI)-1640 medium (Gibco™, Thermo Fisher Scientific) supplemented with 10% heat-inactivated fetal bovine serum, 100 IU/mL penicillin-100 μg/mL streptomycin solution, and 2 mM L-glutamine. Cultured cells were maintained in T-75 flasks at 37 °C in a saturated humid air/5% CO_2_ incubator. Every 3–4 days, at confluence, cells were passaged with 0.25% trypsin/ethylenediaminetetraacetic acid (EDTA) (#25200056, Gibco™) and were used between passages 4 and 8 throughout this study.

HCT116-WT and HCT116-VDR/KO cells were exposed to various concentrations (0.001–20 μM) of VitD_3,_ coclaurine and reticuline for different incubation periods (24, 48, and 72 h). Dimethyl sulfoxide (DMSO), the solvent used for the reconstitution of alkaloids and VitD_3_, was added to the cells at 0.02%, with the final concentration corresponding to the highest compound tested. DMSO was used as a negative control [24].

### 2.4. Immunofluorescence Staining

Cells (5.4 ×10^4/^well) were seeded onto a Nunc^®^ Lab-Tek™ II chambered coverglass (Lochhamer Schlag 11, Gräfelfing, Germany). After 24 h of incubation, untreated cells and cells treated with VitD_3_, coclaurine, and reticuline tested at 20 μM were incubated for 72 h. Cells were rinsed with phosphate-buffered saline (PBS) and then fixed for 30 min at room temperature with 4% formaldehyde diluted in PBS. Membrane permeabilization was performed with 0.1% Triton X-100 in PBS for 10 min at room temperature. Cells were washed with PBS and blocked with 2.5% bovine serum albumin for 1 h, followed by washing again with PBS. Cells were then incubated with the mouse primary antibody directed against the VDR (1:50 dilution) overnight at 4 °C. After washing with PBS-Tween 20, another incubation with fluorescein isothiocyanate (FITC)-conjugated mouse IgG2a secondary antibody (#A282470, 1:10 dilution, Antibodies.com Ltd., Cambridge, UK) was applied for 1 h. Cells were stained with Hoechst 33342 for nuclear staining. Immunofluorescence staining was captured using an LSM780 confocal laser scanning microscope (Carl Zeiss Microscopy GmbH, Jena, Germany). Nuclear and cytoplasmic VDR expression levels were quantified based on the mean of fluorescence intensity using ImageJ software version 1.53e (https://imagej.net/ij/index.html, accessed on 2 March 2023) as previously described [25].

### 2.5. Cell Growth Rate

Cells (0.5 × 10^4^/well) were seeded into an opaque 96-well plate (Greiner^®^, Kremsmunster, Austria). After different incubation times (24, 48 and 72 h), cell growth was evaluated using the CellTiter-Glo^®^ Luminescent Cell Viability Assay (Promega, Madison, WI, USA). This assay detects the quantity of ATP produced, a hallmark of cell viability and growth. The kit was used according to the manufacturer’s instructions. Briefly, CellTiter-Glo^®^ reagent (100 µL) was added to each well, mixed on an orbital shaker for 2 min, and then the plate was protected from light for 10 min at room temperature. The luminescent signal was monitored using an EnVision microplate reader (PerkinElmer, Waltham, MA, USA).

### 2.6. Scratch-Wound-Healing Assay

Cells (0.2 × 10^6^/well) were seeded in complete medium in 24-well plates. After 24 h of incubation, cells were scratched straight with a sterile P200 pipette tip as previously described [26]. Cells were washed with 500 μL of PBS to remove unattached cells. After that, 500 μL of fresh medium with or without 0.02% DMSO, 20 μM VitD_3_ and 20 μM alkaloids (coclaurine, reticuline) was added. Photographs of the scratch were taken at 0 and 96 h without treatment and after 48 h of treatment under a microscope at 4× magnification. The gap width analysis was performed with the ImageJ software version 1.53e.

### 2.7. MTT Assay

Cells (0.5 × 10^4^/well) were seeded in 100 μL of complete medium in 96-well plates. After 24 h of incubation, untreated cells and cells treated with 0.02% DMSO (negative control) and various concentrations (0.001–20 μM) of VitD_3_, coclaurine, and reticuline were further incubated for 72 h. Cell viability was assessed using the MTT Cell Proliferation Assay Kit (#M8180) according to the manufacturer’s instructions (Beijing Solarbio Science & Technology Co., Ltd., Beijing, China). Briefly, 3-(4,5-dimthylethiazol-2-yl)-2,5-diphenyltetrazolium bromide (MTT) solution (10 μL, final concentration of 0.5 mg/mL) was added to each well and incubated for 4 h, resulting in violet formazan crystal formation in metabolically active cells. After incubation, the medium was carefully aspirated; 100 μL of DMSO was added to dissolve the crystals, and the plate was shaken for 15 min at room temperature. A microplate spectrophotometer (SpectraMax Plus 384, Molecular Device, LLC., San Jose, CA, USA) was used to read the absorbance at 570 nm. From the concentration-response curves, half-maximal inhibitory concentration (IC_50_) values were determined.

### 2.8. Flow Cytometry

Apoptosis status was assessed using the FITC Annexin V Apoptosis Detection Kit with propidium iodide (PI, #640914, BioLegend, San Diego, CA, USA). HCT116-WT and HCT116-VDR/KO cells were treated with 0.02% DMSO (negative control), 20 μM of VitD_3_, coclaurine, and reticuline for 72 h. Briefly, the cells were trypsinized, collected, via centrifugation at 300× *g* for 5 min, washed with PBS, and suspended in 1× binding buffer (100 µL). Then, Annexin V and PI solutions (5 µL) were added and incubated for 15 min in the dark. The cells (10,000) were then analyzed on a FACScanto II flow cytometry system (Becton Dickinson (BD) Biosciences, Franklin Lakes, NJ, USA) using the Diva software v9.0, where no Annexin V detection/no PI detection, Annexin V detection/no PI detection, Annexin V detection/PI detection, and no Annexin V detection/PI detection, indicate viable, early apoptotic, late apoptotic, and necrotic cells, respectively.

The cell cycle was analyzed using the BD Cycletest Plus DNA Reagent Kit (#340242, BD Biosciences) according to the manufacturer’s instructions. Briefly, WT and VDR/KO HCT116 cells were collected after 72 h of incubation with 0.02% DMSO, 20 μM of VitD_3_, coclaurine, and reticuline. After incubation, cells were collected and washed three times with the provided buffered solution. Next, the cells were fixed in the solution A (250 µL), and left at room temperature for 10 min. Then, the cell membrane was permeabilized by adding the solution B (200 µL) and the cells were left at room temperature for 10 min. Finally, the solution C (200 µL) was added to stain the DNA, and the cells were left on ice in the dark for 10 min. The cellular samples (10,000) were then analyzed on a FACScanto II flow cytometer (BD Biosciences) using the Diva software v9.0. Cell cycle histograms were generated using the ModFit LT™ software v.6.0 (Verity Software House, Topsham, ME, USA).

### 2.9. Preparation of Cell Lysates and Western Blot Analysis

Cells (1.5 × 10^5^/well) were seeded into 24-well plates for 24 h, and then cells treated with 0.02% DMSO (negative control), 20 μM of VitD_3_, coclaurine, and reticuline, were further incubated for 72 h. Cells were collected and washed with PBS. The Western blot technology, from the cell lysate preparation, protein estimation, and separation on 12% sodium dodecyl sulfate–polyacrylamide gel electrophoresis (SDS-PAGE) to protein electrotransfer onto nitrocellulose membrane, was performed as previously described [27]. The membranes were incubated overnight at 4 °C on a rotary shaker with the following primary antibodies diluted in blocking buffer: anti-VDR (1:500), cleaved caspase-3 (dilution 1:1000), pro-caspase 3 (1:1000), cleaved PARP (1:500), PARP (1:500), Bcl-2 (1:1000), Bax (1:1000), cyclin A1 (1:1000), cyclin B1 (1:1000), cyclin D1 (1:1000), and anti-GAPDH monoclonal antibodies. After that, the probed membranes were washed three times in TBS-T (Tris-buffered saline, 0.1% Tween-20, pH 7.4) for 10 min at room temperature. The membranes were then stained with infrared fluorescent IRDye^®^ 800RD-conjugated goat anti-mouse and IRDye^®^ 680RD-conjugated goat anti-rabbit secondary antibodies (1:10,000, LI-COR Biosciences, Lincoln, NE, USA) for 1 h at room temperature with continuous mixing. After three washes in TBS-T solution, proteins were visualized using Odyssey^®^ CLx Imaging System (LI-COR Biosciences). Protein expression levels were quantified using the ImageJ software version 1.53e.

### 2.10. RNA Extraction and Reverse Transcription-Quantitative Polymerase Chain Reaction (RT-qPCR)

Cells (1.5 × 10^5^/well) were cultured in 24-well plates. After a 24 h of incubation, untreated cells and cells treated with 20 μM VitD_3_, coclaurine, and reticuline were further incubated for 72 h. Cells were collected and washed twice with PBS. RNA was isolated using the RNeasy Mini Kit (QIAGEN, Hilden, Germany) following the manufacturer’s instructions. Complementary DNAs (cDNAs) were produced from the total RNA extracts through reverse transcription using the Transcriptor first-strand cDNA synthesis kit, and the reaction was performed in a Tetrad2 Thermal Cycler. The targeted genes (i.e., *VDR*, *SNAIL1*, *SNAIL2*, *TP53*, and *ACTB*) and the primer sequences employed are summarized in Table 1. RT-qPCR was performed using a QuantiTect Reverse Transcription kit (Applied Biosystems, Thermo Fisher Scientific) containing the PCR SYBR Green Master Mix and on the Applied Biosystems™ QuantStudio 6 Flex system (Waltham, MA, USA). The related gene expression level was calculated based on the 2^−ΔΔCt^ method [28].

### 2.11. Molecular Docking

The binding interactions of coclaurine and reticuline with the VDR were analyzed after obtaining the 3D crystal structure of VitD (PDB ID: 1DB1) from the RCSB Protein Data Bank [29]. Protein structures were subjected to refinement, minimization, and optimization using the OPLS4 force field via the Protein Preparation Wizard tool (PrepWizard, Schrödinger Release 2024: Glide, Schrödinger, LLC, New York, NY, USA, 2024). The chemical structures of coclaurine and reticuline were prepared using Schrödinger’s LigPrep tool, which generated several optimized and minimized conformations at their lowest energy states. Molecular docking was performed using GLIDE with standard precision (SP) and extra precision (XP) scoring functions. A post-docking analysis was conducted to thoroughly evaluate the docked poses [30,31,32].

### 2.12. Statistical Analysis

The results are expressed as the mean ± standard deviation (SD) from three independent experiments. An unpaired two-tailed Student *t*-test was used for comparison between the two groups. One-way ANOVA followed by a post hoc Tukey test was used to determine the statistical significance of multiple group comparisons. Values of *p* < 0.05 were considered significant.

## 3. Results

### 3.1. Loss of VDR Expression Level in CRC HCT116-VDR/KO Cells and Its Functional Impact

The expression levels of different VDR protein isoforms in HCT116-WT and CRISPR/Cas9-edited_HCT116-VDR/KO cells were assessed using Western blotting technology. Figure 1A shows the quantification of three protein isoforms, which were distinguished by their molecular weights: 60 kDa, 54 kDa, and 48 kDa. In HCT116-WT cells, VDR isoforms were clearly visualized, with the 60 kDa isoform being the most abundant, indicating that HCT116-WT cells expressed significant levels of VDR protein isoforms. In HCT116-VDR/KO cells, the expression of all VDR isoforms was drastically reduced compared with HCT116-WT cells, especially the 54 kDa VDR isoform (Figure 1A). The gene expression level of *VDR* was assessed using RT-qPCR in HCT116-WT and HCT116-VDR/KO cells. The results showed that the *VDR* gene was highly expressed in HCT116-WT cells, while its expression level was weakly monitored in HCT116-VDR/KO cells (Figure 1B). VDR expression and its cellular localization were visualized in HCT116-WT and HCT116-VDR/KO cells using confocal fluorescence microscopy. The VDR (green fluorescence) appeared highly expressed and was detected within the nucleus (blue fluorescence) and in the cytoplasm, depicting a perinuclear localization (Figure 1C). In HCT116-VDR/KO cells, VDR expression was significantly reduced, confirming the dramatic decrease in VDR expression (Figure 1C).

The HCT116-VDR/KO cell line was functionally studied by performing scratch-wound-healing and cell growth assays. The scratch-wound-healing assay was assessed by monitoring the shrinkage and closure of damaged wound areas generated after the scratching of HCT116-WT and HCT116-VDR/KO cell monolayers over time. As shown in representative photomicrographs, after 96 h of incubation, the gap size in the wounded HCT116-WT cell monolayer was larger than that measured in the HCT116-VDR/KO cell monolayer, demonstrating that HCT116-VDR/KO cells exhibited significantly faster wound-healing than HCT116-WT cells. These results confirmed that low VDR expression promotes CRC cell motility and the wound-healing process (Figure 1D). The growth rate of HCT116-WT and HCT116-VDR/KO cells was assessed using the CellTiter-Glo^®^ method at different incubation times (24, 48, and 72 h). At 24 h, both cell lines exhibited similar growth rates. However, after 48 h, HCT116-VDR/KO cells exhibited a slight increase in cell growth compared with HCT116-WT cells, indicating that augmented cell growth may be linked to low VDR expression. This trend continued at 72 h, with HCT116-VDR/KO cells retaining higher cell numbers than HCT116-WT cells (Figure 1E).

### 3.2. Coclaurine and Reticuline Decrease the CRC Cell Viability Through VDR

Viability was assessed after treatment of HCT116-WT and HCT116-VDR/KO cells with various concentrations (0.001–20 μM) of coclaurine, reticuline, and VitD_3_ for 72 h using the colorimetric MTT assay. Compared to the high cell viability assessed in untreated (control, normalized to 100%) and DMSO-treated cells, VitD_3_ significantly decreased HCT116-WT cell viability in a dose-dependent manner by 20% (*p* < 0.01) when tested at 0.001 μM and by 58.5% (*p* < 0.0001) when tested at 20 μM (Figure 2A). At increasing concentrations, like VitD_3_ (Figure 2A), coclaurine and reticuline significantly and dose-dependently inhibited the viability of HCT116-WT cells, resulting in a decrease of 61.5% (*p* < 0.0001, Figure 2B) and 58.7% (*p* < 0.0001, Figure 2C) when tested at 20 μM, while the viability of HCT116-VDR/KO cells was not affected at any concentrations tested, compared to the control (Figure 2). The IC_50_ values of coclaurine, reticuline, and VitD_3_ for HCT116-WT cell viability were 26.2 μM, 17.1 μM, and 15.7 μM, respectively. For further investigation, all compounds were tested at 20 μM, concentration around the IC_50_ values (Table 2).

### 3.3. Coclaurine and Reticuline Increase Nuclear VDR Expression, Upregulate VDR and TP53, and Downregulate SNAIL in HCT116-WT Cells but Not in HCT116-VDR/KO Cells

In untreated HCT116-WT cells, the VDR (green fluorescence) was distributed primarily in the cytoplasm, with minimal nuclear localization (Figure 3). The addition of VitD_3_, the main ligand of VDR, resulted in increased VDR expression, which was accompanied by significant nuclear localization, compared with untreated cells (Figure 3). In HCT116-WT cells exposed to coclaurine, VDR signal intensity increased moderately in the nucleus and cytoplasm, with punctate localization compared with untreated cells, but not as prominently as with VitD_3_. In HCT116-WT cells treated with reticuline, VDR expression in the cytoplasm was slightly higher than in untreated cells, but it was less pronounced than that upon treatment with VitD_3_ (Figure 3). Additionally, pronounced nuclear localization of the VDR was observed in reticuline-treated HCT116-WT cells. Unlike HCT116-WT cells, untreated HCT116-VDR/KO cells exhibited minimal green fluorescence in the cytoplasm and nucleus, indicating low overall VDR expression (Figure 3). No noticeable increase in VDR expression or nuclear translocation was observed in any of the treated (VitD_3_, coclaurine, and reticuline) HCT116-VDR/KO cells, compared with untreated cells (Figure 3).

The gene expression levels of *VDR* and the VDR target genes were monitored using RT-qPCR technology in RNA extracts isolated from untreated (control) HCT116-WT and HCT116-VDR/KO cells and cells treated with VitD_3_, coclaurine, and reticuline. Like VitD_3_, coclaurine and reticuline significantly increased the *VDR* gene expression level in treated HCT116-WT cells, compared with untreated cells (Figure 4A). It is noteworthy that, upon coclaurine treatment, *VDR* gene expression was upregulated to the lowest extent, while upon reticuline treatment, the upregulated *VDR* reached the highest expression level (Figure 4A). Furthermore, as in VitD_3_-treated HCT116-WT cells, there was a significant decrease in the expression level of the oncogenic genes *SNAIL1* and *SNAIL2,* while the upregulation of the tumor suppressor *TP53* was observed in coclaurine- and reticuline-treated HCT116-WT cells (Figure 4A). In HCT116-VDR/KO cells, no significant effects were observed on the gene expression levels of *VDR*, *SNAIL1*, *SNAIL2*, or *TP53* under any of the experimental conditions (Figure 4B).

### 3.4. Coclaurine and Reticuline Reduce the CRC Wound-Healing Process Through VDR

A scratch-wound-healing assay was performed to determine the effects of VitD_3_, coclaurine and reticuline on the wound-healing process using HCT116-WT and HCT116-VDR/KO cells. As shown in Figure 4, after 48 h of incubation, VitD_3_, coclaurine, and reticuline significantly reduced the wound-healing process capacity of HCT116-WT cells compared with the control (untreated) and negative control (DMSO-treated) cells. In contrast, no significant effect was observed on the wound-healing process when using HCT116-VDR/KO cells after treatment with VitD_3_, coclaurine, and reticuline compared with the control and DMSO-treated cells (Figure 5).

### 3.5. Coclaurine and Reticuline Induce Late Apoptosis and Modulate Apoptosis-Related Proteins in CRC Cells Through VDR

The apoptotic status was determined using fluorescence-activated cell sorting (FACS) analysis following Annexin V/PI double staining to investigate the antiproliferative effect of VitD_3_, coclaurine, and reticuline in HCT116-WT and HCT116-VDR/KO cells. Representative scatter plots show that, compared with the negative control (highly viable DMSO-treated cells), HCT116-WT cells exposed to 20 μM of VitD_3_, coclaurine, and reticuline underwent late apoptosis (Figure 6A). HCT116-WT cells exposed to VitD_3_ resulted in an increase in the percentage of late apoptotic cells by ~30% (*p* < 0.05) compared with the DMSO-treated cells (Figure 6A). Coclaurine led to an increase in the percentage of late apoptotic cells (19%, *p* < 0.05) compared with DMSO (Figure 6A). The apoptotic cell percentage after reticuline treatment was increased by 17% (*p* < 0.05) compared with the DMSO-treated cells (Figure 6A). In contrast, there was no significant effect on the percentage of apoptotic cells in HCT116-VDR/KO cells after treatment with VitD_3_, coclaurine, and reticuline compared with DMSO (Figure 6B).

To confirm the induction of apoptosis, the protein expression level of the key enzymes and mitochondrial proteins that are well known to promote apoptosis was assessed using Western blot technology. Thus, the occurrence of apoptosis was verified by detecting the expression levels of cleaved caspase-3, cleaved PARP, Bax, and Bcl-2. The results showed a clear detection of these apoptotic markers in HCT116-WT cells treated with 20 µM of VitD_3_, coclaurine, and reticuline for 72 h, but not in HCT116-VDR/KO cells (Figure 6C). A significant increase in cleaved caspase-3 expression was detected the most in coclaurine-treated HCT116-WT cells and the least in reticuline-treated HCT116-WT cells, while cleaved PARP was significantly increased in HCT116-WT cells treated with VitD_3_ and reticuline (Figure 6D). Additionally, Bax expression was significantly augmented in VitD_3_- and coclaurine-treated HCT116-WT cells, while Bcl-2 expression significantly decreased in reticuline-treated HCT116-WT cells (Figure 6D). In HCT116-VDR/KO cells, there was no significant effect on cleaved caspase-3, cleaved PARP, or Bcl-2 across treatments, except for Bax, whose expression was significantly increased upon coclaurine treatment (Figure 6D).

### 3.6. Coclaurine and Reticuline Cause CRC Cell Growth Arrest in the S-Phase Through VDR

In addition to apoptosis, cell cycle distribution was determined in HCT116-WT and HCT116-VDR/KO cells exposed to 20 μM of VitD_3_, coclaurine, and reticuline, followed by DNA staining with PI and monitoring of DNA content in the percentage of cells using the ModFit LT™ software for flow-cytometric analysis. HCT116-WT cells treated with DMSO showed a cell cycle distribution, with approximately 50–55% of the cell population in G0/G1, 45% of cells in the S-phase, and 7–8% of cells in the G2/M-phase (Figure 7A). Coclaurine treatment resulted in a decrease in the percentage of HCT116-WT cells in the G0/G1-phase to 40% and an increase in the cell percentage in the S-phase, reaching 48% (Figure 7A). Additionally, coclaurine-treated HCT116-WT cells showed an augmentation of the cell percentage in the G2/M-phase up to 12% (Figure 7A). The treatment of HCT116-WT cells with reticuline reduced the percentage of cells (35%) in the G0/G1-phase, with a concomitant increase in the percentage of cells in the S-phase reaching 50% of the cell population, followed by an increase of about 15% of cells in the G2/M-phase (Figure 7A). In contrast, there was no change observed in the percentage of cells in the G0/G1-, S-, and G2/M-phases in HCT116-VDR/KO cells after treatment with VitD_3_, coclaurine, and reticuline (Figure 7B). Using Western blot analysis, a significant increase in the expression levels of cyclins A1 and B1 was observed in coclaurine and reticuline-treated HCT116-WT cells compared with DMSO-treated cells (Figure 7C). In contrast, no significant effect was observed in any of the treated HCT116-VDR/KO cells (VitD_3_, coclaurine, and reticuline) compared with DMSO (Figure 7C).

### 3.7. Molecular Docking Reveals Binding Interactions Between Coclaurine, Reticuline, and VDR

After demonstrating the crucial role of the VDR in the anticancer and pro-apoptotic activities of coclaurine and reticuline, the molecular docking analysis of both alkaloids with the VDR revealed significant insights into their binding interactions. The 3D crystal structure of VitD (PDB ID: 1DB1) was used, and the protein structures were refined, minimized, and optimized using the OPLS4 force field. The VitD_3_ structure showed two interactions, forming hydrogen bonds with Tyr-143 and Ser-278 (Figure 8). The docking score for VitD_3_ was −11.598. As shown in Figure 8A, coclaurine exhibited four interactions in the docking analysis: hydrogen bonds formation with Ser-237 and Tyr-143 and aromatic hydrogen bond interactions with Ser-278 and Tyr-143. The docking score for coclaurine was −8.668. Although coclaurine occupied the same region as VitD_3_, its shorter structure prevented hydrogen bonding with distant residues, resulting in a lower docking score than that of VitD_3_ (Figure 8A). Reticuline showed two interactions in the docking analysis, forming hydrogen bonds with Ser-237 and His 397 (Figure 8B). The docking score for reticuline was −8.556. Similarly to coclaurine, reticuline occupied the same binding region within the VDR as VitD_3_ but had a shorter structure, limiting its ability to form hydrogen bonds with distant residues (Figure 8B). The interactions and docking score of reticuline were comparable to those of coclaurine. Overall, these molecular docking predictions reveal the occupancy of the VDR’s active site by coclaurine and reticuline, confirming the crucial role of the VDR in the anticancer and pro-apoptotic activities of both alkaloids observed using HCT116-WT and HCT116-VDR/KO cells.

## 4. Discussion

Research on phytomedicine, such as alkaloids from *A. muricata*, has attracted considerable interest in clinical practice, particularly for the treatment of cancers such as CRC [33,34,35]. The alkaloids coclaurine and reticuline are widely distributed in plants and have been suggested to exhibit potential anti-CRC activities through growth factor receptor blockade [21]. Besides growth factor receptors, the VDR and its ligands have been described for their oncoprotective actions through the suppression of oncogenic target genes [36]. Loss of VDR expression is well known to promote tumor development and progression [36,37]. Thus, targeting the VDR, resulting in overexpression and activation, could be a therapeutic strategy for cancer management [13].

The VDR, a ligand-activated transcription factor and a relevant prognostic marker for CRC patients, forms a heterocomplex with its main ligand VitD_3_, which subsequently prevents CRC and even inhibits CRC development and progression [7,36]. When activated by its ligand, the VDR exhibits a perinuclear localization, is then translocated into the nucleus, and subsequently binds to target genes for expression, making its cellular localization an indicator of its activity [38]. In the present study, to investigate the impact of low VDR expression levels in CRC cells, the HCT116-VDR/KO cell line was generated using CRISPR/Cas9 technology. VDR loss was verified by detecting low expression using Western blot analysis, an RT-qPCR assay, and immunofluorescence staining. At the functional level, effective VDR gene KO was revealed by an increase in the CRC wound-healing process and growth rate. These results corroborate epidemiological studies reporting that decreased VDR expression levels detected in CRC tissues (compared with adjacent tissues) and in serum (compared with healthy subjects) were associated with CRC progression and poor survival in CRC patients [7,39].

Furthermore, in the present study, at the level of VDR target gene expression, coclaurine and reticuline significantly upregulated *VDR* and tumor suppressor *TP53* expression while markedly downregulating *SNAIL1* and *SNAIL2* in HCT116-WT cells. However, these compounds lost their modulatory effects on the expression of the *VDR*, *SNAIL1*, *SNAIL2*, and *TP53* in HCT116-VDR/KO cells due to the loss of the VDR. SNAIL1 and SNAIL2 are zinc-finger transcription factors that have been shown to inhibit VDR expression and play a crucial role in CRC development and progression through the induction of the protein-related epithelial–mesenchymal transition (EMT) process, leading to metastasis and contributing to drug resistance [40,41]. Hence, both SNAIL1 and SNAIL2 have been reported as good prognostic markers for CRC patients when weakly expressed and become promising targets for the development of novel therapeutic strategies [41,42,43]. The repression of *SNAIL1* and *SNAIL2* gene expression by coclaurine and reticuline provides novel insights into the possible ability of alkaloids to target these key transcription factors involved in the EMT process. In addition, VitD_3_-induced *TP53* expression through the VDR has been widely reported in several types of cancers [36,44], which supports the anticancer effects of coclaurine and reticuline by upregulating *TP53* gene expression.

Like VitD_3_, coclaurine and reticuline dramatically inhibited the wound-healing process in HCT116-WT cells. However, no effect was observed in HCT116-VDR/KO cells, indicating the key role of the VDR in both alkaloids’ impact on CRC cell motility and proliferation, the two main cellular events involved. Berbamine, a bisbenzylisoquinoline alkaloid, has previously been shown to inhibit the migration and growth of human CRC cell lines [45]. Moreover, the VDR was demonstrated to be responsible for decreasing CRC cell migration by inhibiting Wnt/β-catenin signaling, a pathway that stimulates EMT and that is promoted by SNAIL [36,42,46]. Thus, further investigation into the molecular mechanisms by which coclaurine and reticuline inhibit CRC cell motility focusing on VDR downstream effectors, particularly the Wnt/β-catenin pathway, would be of interest for the discovery of potential therapeutic targets. In the current study, coclaurine and reticuline, like VitD_3_, decreased CRC cell viability in HCT116-WT cells in a dose-dependent manner, but not in HCT116-VDR/KO cells. The cytotoxic effect of alkaloids extracted from *Peganum harmala* seeds against HCT116 cells was also recently reported [47]. The loss of the antiproliferative effects of coclaurine and reticuline on HCT116-VDR/KO cells indicates the crucial presence of the VDR and its activity for these compounds’ ability to promote cell death. Further in vitro research is needed to explore the molecular mechanisms underlying coclaurine- and reticuline-induced cell death in CRC cells.

Cell death is controlled by the balance between pro-apoptotic proteins and anti-apoptotic proteins. The apoptotic status promoted by coclaurine and reticuline was revealed by the increased detection of late apoptotic cells, which is associated with the increased expression of key pro-apoptotic protein markers, including cleaved caspase-3, an executioner caspase, and cleaved PARP, a hallmark of the intrinsic pathway involved in oligonucleosomal DNA fragmentation, as well as the decreased expression of the key anti-apoptotic protein marker Bcl-2 [48,49]. All of these hallmarks of apoptosis were significantly modulated towards apoptosis in HCT116-WT cells treated with coclaurine and reticuline compared with HCT116-VDR/KO cells, in which no effect was observed. Generally, in CRC cells, high expression of *BAX* and *TP53* induces apoptosis, while elevated levels of *BCL-2* and *BCL-xL* inhibit cell death [50,51,52]. Consistent with this framework, coclaurine and reticuline upregulated the pro-apoptotic protein Bax and downregulated the anti-apoptotic gene *BCL-2* in HCT116-WT cells, demonstrating the key role of the VDR in the pro-apoptotic effects of these alkaloids in CRC cells. Other alkaloids, such as lycorine and its hydrochloride derivative from *Amaryllidaceae* plants, also induce intrinsic apoptosis by altering mitochondrial membrane potential (MMP) via the modulation of the *BCL-2* protein family [49,53,54,55]. This process activates the caspase cascade, resulting in increased levels of cleaved caspase-3, cleaved caspase-9, and cleaved PARP, culminating in apoptotic cell death [49,53,54,55,56]. Kim and colleagues demonstrated that harmine, a carboline alkaloid from *Peganum harmala*, induced apoptotic cell death in HCT-116 cells by elevating pro-apoptotic markers (caspases 3 and 9, PARP, and Bax), while reducing anti-apoptotic protein *BCL-2* levels [48]. The findings indicated that the harmine hydrochloride molecule primarily modifies the ERK/PI3K/Akt/mTOR signaling pathway to induce apoptosis [48]. Further investigations into the pro-apoptotic effects of coclaurine and reticuline in CRC cells on MMP alteration and signaling pathway modulation would be of great interest.

Moreover, upon activation by vitamin D, the VDR regulates the expression of apoptosis-related genes, inhibiting anti-apoptotic proteins (e.g., BCL2 and BCL-XL) and enhancing pro-apoptotic factors (e.g., BAX, BAK, and BAD), thus facilitating programmed cell death [57]. A previous study revealed that the knockdown of the VDR attenuated the antiproliferative, pro-apoptotic, and anti-invasive effects of vitamin D in papillary thyroid cancer (PTC) cells, potentially through the activation of the Wnt/β-catenin signaling pathway. The findings of this study indicate that the VDR could serve as a unique and potential therapeutic target for the treatment of PTC [44]. Furthermore, it is noteworthy that VDR overexpression in human CRC decreases the expression of β-catenin and facilitates its nuclear exportation to the cytoplasm, subsequently inhibiting Wnt/β-catenin signaling, which leads to reduced lymphoid enhancer-binding factor 1 (LEF1) levels and, consequently, a reduction in the expression of cyclin D1, which is likely to increase cancer cells’ sensitivity to apoptosis [36]. The VDR-dependent pro-apoptotic effects of coclaurine and reticuline require in-depth investigation focusing on other proteins leading to cell death, such as cyclins causing cell growth arrest.

In addition to the induction of apoptosis, anticancer therapies aim to induce cell cycle arrest [58,59]. The cell cycle consists of distinct phases (G0/G1, S, and G2/M) that regulate cell growth, division, and proliferation [58]. Tight regulation of this process depends on cyclins, cyclin-dependent kinases (CDKs), and CDK inhibitors (CDKIs) [58]. Specifically, cyclin D1 controls the G0/G1-phase, while cyclin B1 regulates the G2/M-phase [45,60]. Cyclin A1, which is produced during the G1/S transition, promotes progression into the S-phase but is degraded by the G2/M-phase [61,62]. In the current investigation, flow cytometry analysis indicated S-phase arrest in HCT116-WT cells following treatment with coclaurine and reticuline, but no significant alteration was noted in HCT116-VDR/KO cells. The S-phase arrest was associated with elevated expression of cyclins A1 and B1, indicating that the VDR is involved in modulating cell cycle progression in response to these alkaloids. Our findings align with those about lycorine and its derivative, which were shown to induce S-phase arrest in the MKN-45 and SGC7901 gastric cancer cell lines, resulting in the accumulation of DNA damage and subsequent apoptosis [55]. Activation of the VDR by vitamin D or other analogs causes cell cycle arrest in the G1/G0-phase for cancer cells. This is mainly accomplished through the induction of CDKIs p21 and p27, which block the activities of cyclins D and E [63]. Further in-depth studies on the activities of CDKs and CDKIs modulated by coclaurine and reticuline in CRC cells would provide more insights into the molecular mechanisms involved in cell growth arrest.

An in silico molecular docking study was performed to reveal the existence of any binding interactions between the alkaloids coclaurine and reticuline and the VDR. The docking results indicated higher binding energy and stability of VitD_3_ for the VDR than the new ligands, coclaurine and reticuline. Coclaurine and reticuline showed multiple interactions, such as hydrogen bonds and aromatic hydrogen bonds, and binding activity at the VDR. However, unlike VitD_3_, both alkaloids exhibited a much smaller structure that restricted hydrogen bonding to proximal residues, leading to lower binding affinities than that of VitD_3_. Although these molecular docking results provide valuable insights into the interactions between alkaloids (i.e., coclaurine and reticuline) and the VDR, it is crucial to consider that at the functional level, the anticancer effects of coclaurine and reticuline were found to be as effective as VitD_3_ and even stronger with respect to reticuline-induced VDR upregulation. Further biological evidence demonstrating the degree of binding of these alkaloids with the VDR relative to VitD_3_ is warranted, such as competitive binding assays. For potential therapeutic application, the pharmacokinetics, organ toxicity (i.e., neurotoxicity, cardiotoxicity, nephrotoxicity, hepatotoxicity), safety, and bioavailability of coclaurine and reticuline need to be evaluated in vivo.

## 5. Conclusions

In this study, the findings provide compelling evidence for the potential of coclaurine and reticuline as novel therapeutic agents in CRC treatment. Our results, summarized in the schematic diagram (Figure 9), demonstrate that these alkaloids exhibit significant anticancer activity through the upregulation of *VDR* and *TP53* and pro-apoptotic effects in a VDR-dependent manner. Although molecular docking studies revealed that coclaurine and reticuline bind to the VDR through interactions that are not as strong as those of VitD_3_, reticuline exhibited the strongest effect on VDR upregulation. Further in vivo studies using CRC models with assessments of toxicity, pharmacokinetics, and bioavailability are needed to validate coclaurine and reticuline as potential anti-CRC drugs and inducers of VDR expression.

## Figures and Tables

**Figure 1 cimb-47-00810-f001:**
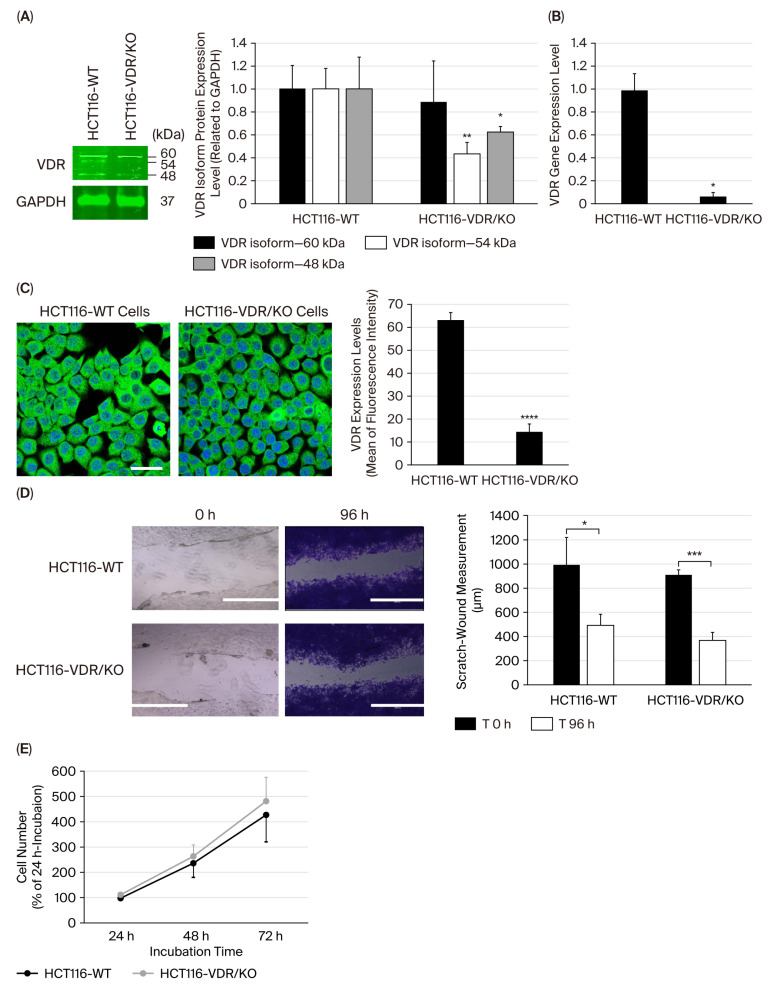
**Detection of VDR expression in HCT116-WT vs. HCT116-VDR/KO cells and evaluation of VDR loss in CRC wound-healing and cell growth.** (**A**) Representative Western blots showing VDR protein expression levels in HCT116-WT cells compared with HCT116-VDR/KO cells. Bar graph of quantitative analysis of VDR protein isoforms related to GAPDH, the loading control. (**B**) RT-qPCR assay showing low *VDR* gene expression detected in HCT116-VDR/KO cells, compared with the high expression level monitored in HCT116-WT cells. (**C**) Representative photomicrographs showing the expression and localization of the VDR, indicated in green fluorescence, in HCT116-WT and HCT116-VDR/KO cells, with their nuclei, indicated in blue fluorescence. Scale bar = 20 μm. Bar graph reporting VDR expression levels based on the mean of fluorescence intensity measured in six random fields of HCT116-WT and HCT116-VDR/KO cell monolayers using ImageJ software. (**D**) Representative photomicrographs showing the wounded area at time 0 h and its shrinkage at 96 h. Scale bar = 1000 μm. Bar graph summarizing measurements of the shrinkage of the wounded areas in five sections, compared with the measurements taken at the initial time (T 0h), using the ImageJ software. (**E**) The cell growth rate was determined using the CellTiter-Glo^®^ Luminescent Cell Viability Assay. Each experiment was independently repeated three times and presented as the mean ± SD. (*), (**), (***), and (****) signify a statistically significant difference (*p* < 0.05, *p* < 0.01, *p* < 0.001, and *p* < 0.0001) compared with the HCT116-WT cells.

**Figure 2 cimb-47-00810-f002:**
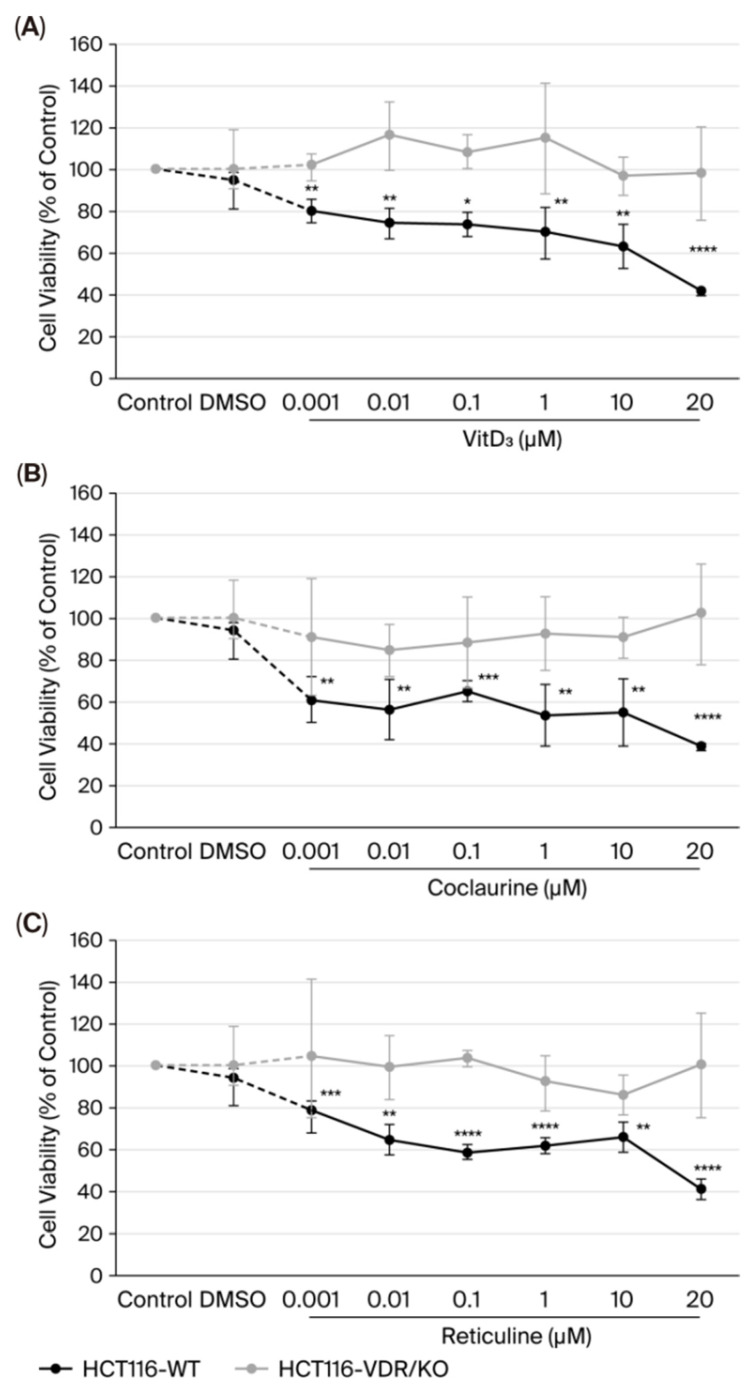
**Assessment of various concentrations (0.001–20 μM) of VitD_3_ (A), coclaurine (B), reticuline (C), and 0.02% DMSO (negative control) on HCT116-WT and HCT116-VDR/KO cell viability after 72 h of incubation using the MTT assay.** The results were normalized to the control (100%) and are presented as the mean ± SD from three independent experiments. (*), (**), (***) and (****) signify a statistically significant difference (*p* < 0.05, *p* < 0.01, *p* < 0.001 and *p* < 0.0001), compared with the control (untreated cells).

**Figure 3 cimb-47-00810-f003:**
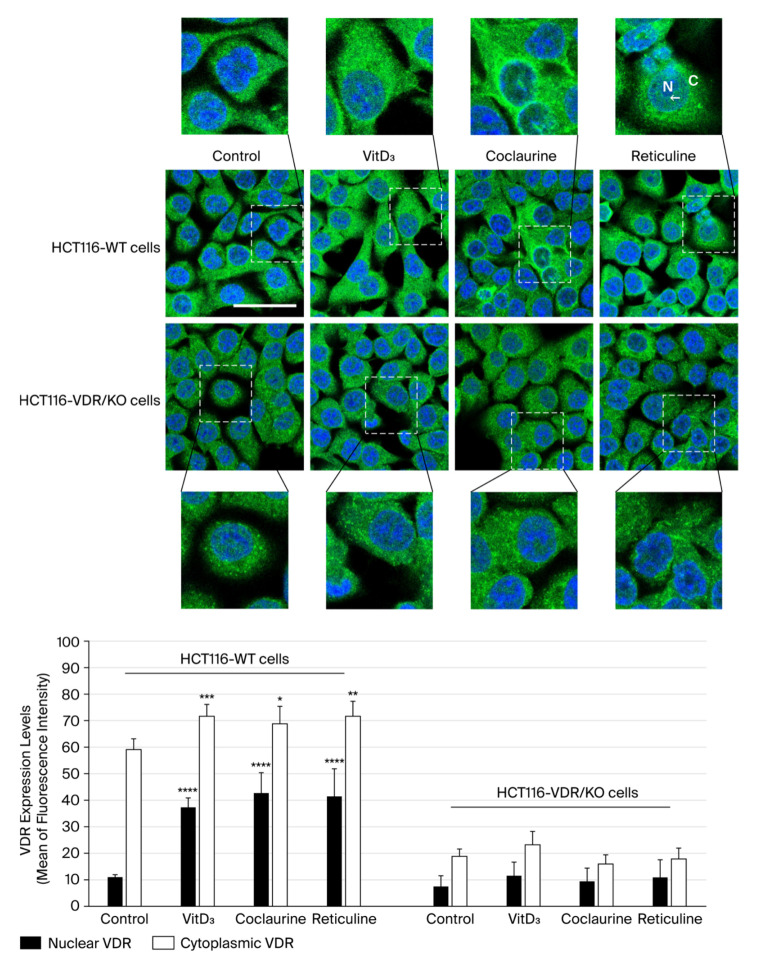
**Representative photomicrographs showing VDR localization in HCT116-WT and HCT116-VDR/KO cells using immunofluorescence staining after 72 h of incubation in the presence or absence of 20 μM of VitD_3_, coclaurine, and reticuline.** Scale bar = 20 μm. The insert from Reticuline-treated cells depicts a higher magnification of an example of a VDR nuclear location that is indicated by an arrow. “C” stands for cytoplasm and “N” stands for nucleus. The bar graph summarizes the mean of fluorescence intensity corresponding to the expression levels of VDR detected nuclear and cytoplasmic localizations in six random fields. The results are presented as the mean ± SD from three independent experiments. (*), (**), (***), (****) signify a statistically significant difference (*p* < 0.05, *p* < 0.01, *p* < 0.001, and *p* < 0.0001) compared with the control (untreated cells).

**Figure 4 cimb-47-00810-f004:**
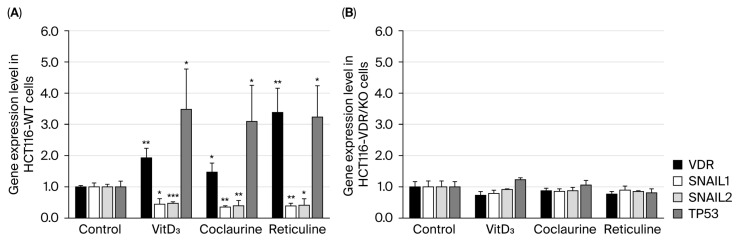
**Gene expression levels of *VDR*, *SNAIL1*, *SNAIL2*, and *TP53* in (A) HCT116-WT and (B) HCT116-VDR/KO cells were assessed using RT-qPCR technology.** Cells were incubated with or without 20 μM VitD_3_, coclaurine and reticuline for 72 h. The experiment was independently repeated three times, and the data were normalized to *β-actin*. The results are presented as the mean ± SD. The (*), (**), and (***) signify a statistically significant difference (*p* < 0.05, *p* < 0.01, and *p* < 0.001) compared with the control (untreated cells).

**Figure 5 cimb-47-00810-f005:**
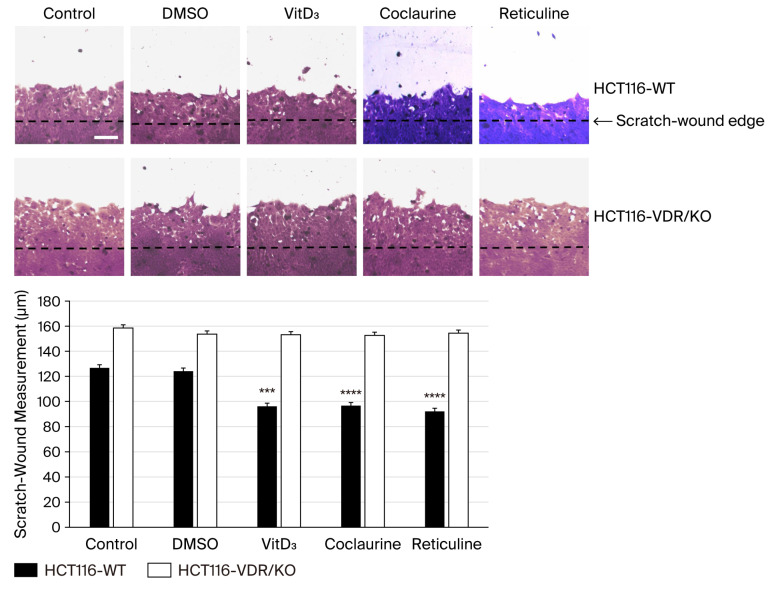
**Evaluation of the effect of VitD_3_, coclaurine, and reticuline on the wound-healing process using HCT116-WT and HCT116-VDR/KO cells by performing a scratch-wound-healing assay.** Representative photomicrographs showing the wound-healing process of untreated cells, DMSO-treated cells and cells treated with 20 μM VitD_3_ and 20 μM alkaloids after 48 h of incubation. Scale bar = 100 μm. The wound recovery distance from the wound edge was measured in five sections using the ImageJ software. Data are presented as the mean ± SD from three independent experiments. (***) and (****) signify a statistically significant difference (*p* < 0.001 and *p* < 0.0001) compared with the control (untreated cells).

**Figure 6 cimb-47-00810-f006:**
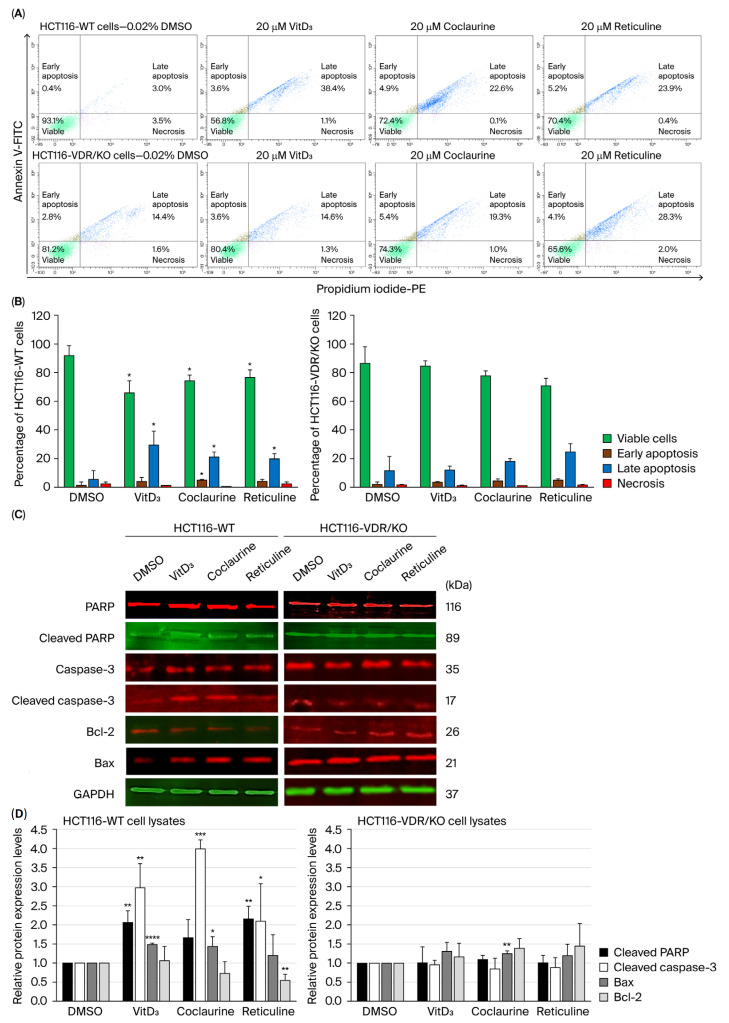
**Apoptotic status and detection of apoptotic proteins in HCT116-WT and HCT116-VDR/KO cells exposed to VitD_3_, coclaurine, and reticuline.** (**A**) Flow cytometry scatter plots of cells treated for 72 h with 0.02% DMSO (negative control), 20 µM of VitD_3_, coclaurine, and reticuline. (**B**) Bar graph of the percentages of HCT116 cell status, determined as viable, early apoptosis, late apoptosis, and necrosis. (**C**) Representative Western blot results showing an increase in the expression levels of cleaved caspase-3, cleaved PARP, and Bax in HCT116-WT cells treated with 20 μM of VitD_3_, coclaurine and reticuline compared with DMSO-treated cells and HCT116-VDR/KO cells. (**D**) Bar graph reporting the quantitative analysis of Western blotting using the ImageJ software. The results are presented as the mean ± SD from three independent experiments. (*), (**), (***) and (****) signify a statistically significant difference (*p* < 0.05, *p* < 0.01, *p* < 0.001 and *p* < 0.0001) compared with the DMSO-treated cells.

**Figure 7 cimb-47-00810-f007:**
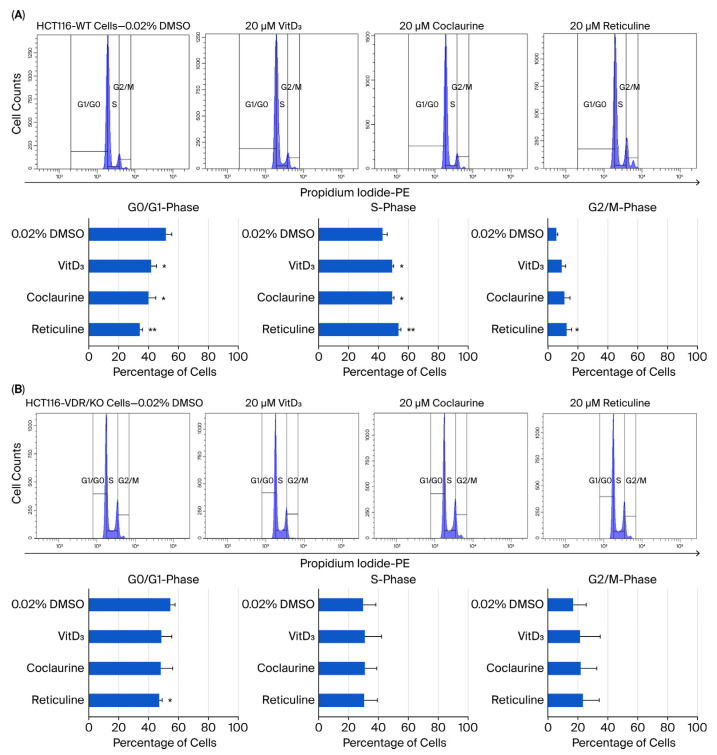
**Cell cycle analysis and cyclin expression levels in HCT116-WT and HCT116-VDR/KO cells after treatment with 0.02% DMSO (negative control), 20 μM of VitD_3_, coclaurine, and reticuline.** Representative flow-cytometric histograms showing the cell cycle distribution in HCT116-WT (**A**) and HCT116-VDR/KO (**B**) cells. Bar graph of the cell cycle analysis for three independent experiments presented as the mean ± SD. The data were normalized to GAPDH and related to the control sample. (**C**) Representative Western blots showing the expression levels of cyclins A1, B1, and D1 in HCT116-WT and HCT116-VDR/KO cell lysates under the indicated experimental conditions. Bar graph reporting the quantitative analysis of Western blotting using the ImageJ software. The experiment was independently repeated three times, and the results are presented as the mean ± SD. The data were normalized to GAPDH and related to the control sample. * *p* < 0.05, ** *p* < 0.01, and *** *p* < 0.001 compared with the DMSO-treated cells.

**Figure 8 cimb-47-00810-f008:**
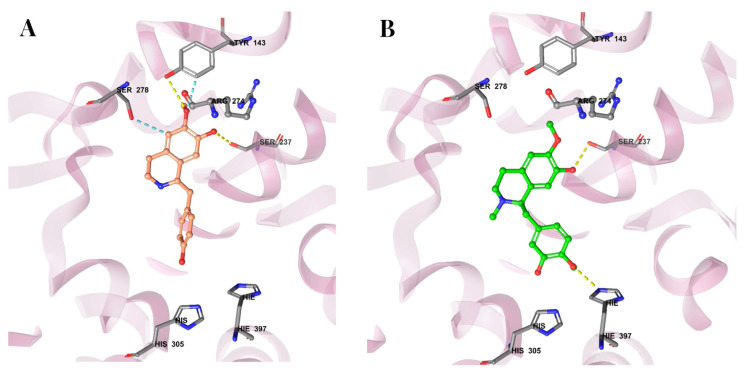
**Molecular docking interactions with the crystal structure of the VDR.** (**A**) Main interactions with the coclaurine structure (in orange). (**B**) Main interactions with the reticuline structure (in green).

**Figure 9 cimb-47-00810-f009:**
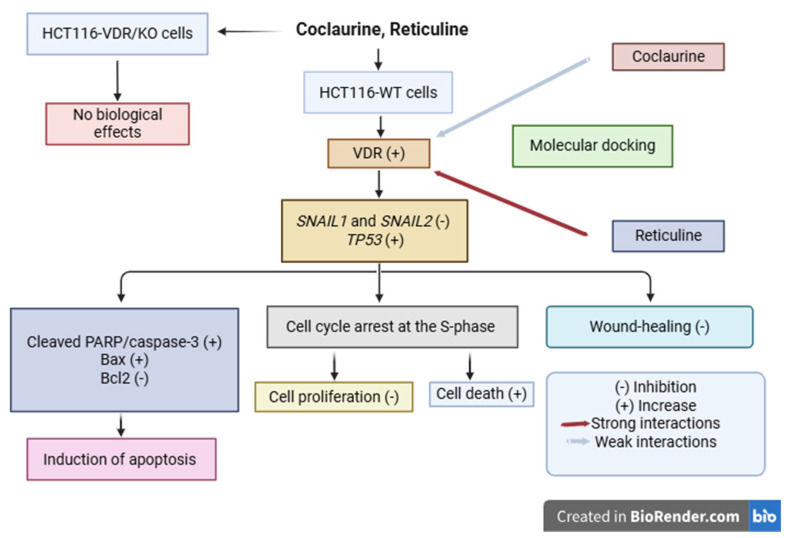
This schematic diagram summarizes the investigated anticancer and pro-apoptotic activities of coclaurine and reticuline targeting VDR using CRISPR/Cas9-edited VDR/KO (HCT116-VDR/KO) and HCT116-WT CRC cell lines. HCT116-WT cells exposed to coclaurine and reticuline led to the upregulation of VDR (resulting in the suppression of its target genes *SNAIL1* and *SNAIL2*) and the tumor suppressor *TP53*, inhibition of the wound-healing process and cell viability. In addition, the cell death mechanisms underlying both alkaloids antiproliferative activities included induction of apoptosis and cell cycle arrest in the S-phase, associated with increased expression of pro-apoptotic proteins (i.e., cleaved PARP, cleaved caspase-3, Bax) and decreased expression of the anti-apoptotic protein (i.e., Bcl-2). However, no biological effects of colcaurine and reticuline were observed in HCT116-VDR/KO cells, demonstrating the crucial role of VDR in both anti-CRC effects of coclaurine and reticuline. Molecular docking study highlighted the strong and weak molecular interactions of reticuline and coclaurine with the active site of VDR, respectively.

**Table 1 cimb-47-00810-t001:** Apoptosis-related gene primer sequences for RT-qPCR and PCR product size in base pairs (bp).

Gene	Forward Primer Sequence (5′-3′)	Reverse Primer Sequence (5′-3′)	PCR Product Size (bp)
*VDR*	CCAGTTCGTGTGAATGATGG	GTCGTCCATGGTGAAGGA	384
*SNAIL1*	GCTCCACAAGCACCAAGAGT	ATTCCATGGCAGTGAGAAGG	145
*SNAIL2*	GAGCATTTGCAGACAGGTCA	GCTTCGGAGTGAAGAAATGC	200
*TP53*	GAGATGTTCCGAGAGCTGAATGAGGC	TCTTGAACATGAGTTTTTTATGGCGGGAGG	1063
*ACTB*	GCTCGTCGTCGACAACGGCTC	CAAACATGATCTGGGTCATCTTCT	352

**Table 2 cimb-47-00810-t002:** IC_50_ values of compounds causing 50% decrease in viability of HCT116-WT cells.

Compounds	IC_50_ Values (μM)
VitD_3_	15.7
Coclaurine	26.2
Reticuline	17.1

## Data Availability

Human CRC expressing VDR wild-type (HCT116-WT) and VDR/KO (HCT116-VDR/KO) cell lines were provided by the American Type Culture Collection (Manassas, VA, USA) via Synthego Corporation (Menlo Park, CA, USA). The original contributions presented in this study are included in the article. Further inquiries can be directed to the corresponding author.

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
