# Peer review of "Reticuline and Coclaurine Exhibit Vitamin D Receptor-Dependent Anticancer and Pro-Apoptotic Activities in the Colorectal Cancer Cell Line HCT116"

_cimb, 2025, doi:10.3390/cimb47100810_

Round 1

Reviewer 1 Report (New Reviewer)

Comments and Suggestions for Authors

The article “Reticuline and coclaurine exhibit vitamin D receptor-dependent anticancer and proapoptotic activities in colorectal cancer cells” addresses an important issue in the search for novel therapeutic strategies against colorectal cancer. The study is carefully and thoroughly prepared, demonstrating the potential significance of plant-derived alkaloids in oncology. Nevertheless, there are several aspects that require clarification and refinement in order for the manuscript to fully meet the standards of publication.

  1. A critical limitation of the study lies in the absence of experiments involving non-malignant (healthy) cell lines as a reference. The assessment of selective cytotoxicity is essential in determining whether the tested compounds truly possess anticancer potential. If reticuline and coclaurine selectively inhibit the proliferation of colorectal cancer cells while sparing normal colon cells, such findings would substantiate their role as promising anticancer agents. However, if both malignant and healthy cells are equally affected, the observed effects should be interpreted as general toxicity rather than cancer-specific activity.
  2. The manuscript does not provide sufficient information regarding the source of the compounds used in the study. It should be clearly stated whether reticuline and coclaurine were obtained through commercial suppliers, isolated from natural sources, or synthesized. This detail is essential for the reproducibility of the experiments and should be explicitly included in the Methods section.
  3. In Table 1, the size of the PCR products should be included. Providing this information is important for clarity and reproducibility, as it allows readers to verify the specificity of the primers and the validity of the experimental design.
  4. In Figure 2, the effect described in the text and reflected in the graph below the images is only weakly visible in the presented photographs. It would be valuable to provide magnified views for all four HCT116-WT images, rather than only for the cells treated with reticuline. Additionally, for the sake of completeness and comparison, it would be highly informative to include a corresponding graph illustrating the results obtained for HCT116-VDR/KO cells.
  5. In the description of several figures, it is not entirely clear whether the concentration of the tested alkaloids was also 20 µM, as in the case of vitamin D. This information should be explicitly stated to avoid ambiguity and to ensure that the experimental conditions are transparent and reproducible.
  6. The subsection of the Methods describing the MTT assay, in which the authors assess cell viability in the presence of the tested compounds, should be placed earlier in the Results section (e.g., as subsection 3.2 rather than 3.4). Since these findings form the basis for selecting the concentrations of the compounds used in subsequent experiments, they logically precede and inform the following analyses.
  7. It would be highly useful to include a table summarizing the IC₅₀ values of the tested compounds. This would provide a clear and concise overview of their cytotoxic potency and facilitate comparison between different cell lines and treatments.
  8. The supplementary material should include the full Western blot. Providing the complete images is essential for transparency, allowing readers to fully assess the specificity and integrity of the detected signals.
  9. The authors correctly emphasize the need for in vivo studies; however, in this context, it would be important to discuss the potential toxicity of these compounds. The experiments were performed using a concentration of 20 µM, but achieving such levels at the target site in vivo would likely require substantially higher doses due to metabolism and distribution. As noted earlier, no experiments were conducted on healthy cells, which limits the assessment of selective cytotoxicity. Additionally, available data indicate that both alkaloids may exhibit neurotoxicity at higher concentrations, a factor that should be carefully considered when discussing future in vivo
  10. The study only examined a single colorectal cancer cell line, HCT116, which was further modified for the experiments. Therefore, the title should clearly reflect that the findings pertain specifically to this cell line. Currently, the title refers more generally to “colorectal cancer cell lines,” which may be misleading, as the sensitivity and molecular characteristics of different colorectal cancer lines can vary substantially. The results obtained with HCT116 cells may not necessarily be reproducible in other lines, such as HT29 or others, highlighting the need for precision in the title and interpretation.

Author Response

RESPONSE TO THE REVIEWER #1

Manuscript ID: CIMB-3878374

Title: Reticuline and coclaurine exhibit vitamin D receptor-dependent anticancer and pro-apoptotic activities in colorectal cancer cell lines

Dear Editor,

Thank you for considering this manuscript for review. We are grateful to all reviewers for their detailed feedback and believe that this has improved the quality of our revised manuscript. Please find below our detailed response (in red) to each point raised by the reviewer #1.

______________________________________________________________________________

The article “Reticuline and coclaurine exhibit vitamin D receptor- dependent anticancer and proapoptotic activities in colorectal cancer cells” addresses an important issue in the search for novel therapeutic strategies against colorectal cancer. The study is carefully and thoroughly prepared, demonstrating the potential significance of plant- derived alkaloids in oncology. Nevertheless, there are several aspects that require clarification and refinement in order for the manuscript to fully meet the standards of publication.

  1. A critical limitation of the study lies in the absence of experiments involving non-malignant (healthy) cell lines as a reference.

The assessment of selective cytotoxicity is essential in determining whether the tested compounds truly possess anticancer potential. If reticuline and coclaurine selectively inhibit the proliferation of colorectal cancer cells while sparing normal colon cells, such findings would substantiate their role as promising anticancer agents. However, if both malignant and healthy cells are equally affected, the observed effects should be interpreted as general toxicity rather than cancer- specific activity.

Thank you for the comments. However, this study aimed to demonstrate the beneficial anticancer effects of these alkaloids through VDR, including promoting VDR expression, using colorectal cancer cells expressing VDR and not expressing VDR. Of course, to check the safety of these alkaloids, normal colon epithelial cells will be used in future with the expectations that these alkaloids will not affect their slow cell growth and high cell viability.

  1. The manuscript does not provide sufficient information regarding the source of the compounds used in the study. It should be clearly stated whether reticuline and coclaurine were obtained through commercial suppliers, isolated from natural sources, or synthesized. This detail is essential for the reproducibility of the experiments and should be explicitly included in the Methods section.

Thank you for this relevant comment. The alkaloids were commercially obtained from Solarbio Science & Technology Co., Ltd. They were extracted from the roots of Annona muricata with a purity ≥98%, analyzed using HPLC. This information has been added to the Materials and Methods, 2.1. Reagents section, as follows:

“Coclaurine (#SC5960) and reticuline (#SR8320) with a purity ≥ 98% were extracted from A. muricata roots and provided by Solarbio Science and Technology Co., Ltd., Beijing, China). ”  

  1. In Table 1, the size of the PCR products should be included.

Providing this information is important for clarity and reproducibility, as it allows readers to verify the specificity of the primers and the validity of the experimental design.

As suggested, this information has been added along to each paired primer sequences in the revised Table 1.

  1. In Figure 2, the effect described in the text and reflected in the graph below the images is only weakly visible in the presented photographs. It would be valuable to provide magnified views for all four HCT116-WT images, rather than only for the cells treated with reticuline. Additionally, for the sake of completeness and comparison, it would be highly informative to include a corresponding graph illustrating the results obtained for HCT116-VDR/KO cells.

Thank you for this comment. We would like to apologize for these figures that eventually became blurry, which made more difficult to see the obvious VDR localizations in WT cells and very low expression in VDR/KO cells. As suggested, we magnified all the high-resolution photomicrographs and enlarged one representative cell for a better observation of VDR expression levels in both cellular compartments (i.e., nucleus and cytoplasm). Statistical analysis was performed for all conditions (WT and VDR/KO cells) after measurement of the mean of fluorescence intensity using imageJ software. A new chart is shown in the corresponding revised Figure (new Figure 3).

  1. In the description of several figures, it is not entirely clear whether the concentration of the tested alkaloids was also 20 µM, as in the case of vitamin D. This information should be explicitly stated to avoid ambiguity and to ensure that the experimental conditions are transparent and reproducible.

Thank you for your comment. We checked and ensured that the concentrations are clearly mentioned in the revised manuscript.

     6.The subsection of the Methods describing the MTT assay, in which the authors assess cell viability in the presence of the tested compounds, should be placed earlier in the Results section (e.g., as subsection 3.2 rather than 3.4). Since these findings form the basis for selecting the concentrations of the compounds used in subsequent experiments, they logically precede and inform the following analyses.

We totally agree with this relevant comment. Therefore, we moved the subsection MTT results to subsection 3.2.

     7.It would be highly useful to include a table summarizing the IC₅₀ values of the tested compounds. This would provide a clear and concise overview of their cytotoxic potency and facilitate comparison between different cell lines and treatments.

As suggested, a table providing IC50 values has been added as Table 2.

     8.The supplementary material should include the full Western blot.

Providing the complete images is essential for transparency, allowing readers to fully assess the specificity and integrity of the detected signals.

All the uncropped Western blots were already provided as unpublished supplementary file. They are  now provided as a supplementary material.

  1. The authors correctly emphasize the need for in vivo studies; however, in this context, it would be important to discuss the potential toxicity of these compounds. The experiments were performed using a concentration of 20 µM, but achieving such levels at the target site in vivo would likely require substantially higher doses due to metabolism and distribution. As noted earlier, no experiments were conducted on healthy cells, which limits the assessment of selective cytotoxicity.

Additionally, available data indicate that both alkaloids may exhibit neurotoxicity at higher concentrations, a factor that should be carefully considered when discussing future in vivo

Thank you for this insightful comment. However, in the end of the Discussion, the toxicity mentioned was referring to organ toxicity. Therefore, we revised the sentence as follows:

“For potential therapeutic application, the pharmacokinetics, toxicity (i.e., neurotoxicity, cardiotoxicity, nephrotoxicity, hepatotoxicity), safety, and bioavailability of coclaurine and reticuline need to be evaluated in vivo.”

  1. The study only examined a single colorectal cancer cell line, HCT116, which was further modified for the experiments. Therefore, the title should clearly reflect that the findings pertain specifically to this cell line. Currently, the title refers more generally to “colorectal cancer cell lines,” which may be misleading, as the sensitivity and molecular characteristics of different colorectal cancer lines can vary substantially. The results obtained with HCT116 cells may not necessarily be reproducible in other lines, such as HT29 or others, highlighting the need for precision in the title and interpretation.

Thank you for this relevant comment. The revised Title is “Reticuline and coclaurine exhibit vitamin D receptor-dependent anticancer and pro-apoptotic activities in the colorectal cancer cell line HCT116”.

We hope these changes are satisfactory as we are keen to publish in Current Issues in Molecular Biology journal. Please feel free to contact us should you require further clarification.

We are looking forward to hearing from you

With my best regards

Sabine Matou-Nasri, PhD.

Senior research scientist

Blood and Cancer Research Department

Affiliate Professor, Biosciences Department, Faculty of the School of Systems Biology, George Mason University, VA, USA

King Abdullah International Medical Research Center (KAIMRC)

Ministry of National Guard-Health Affairs (MNG-HA)

P O Box 22490; Mail Code# 1515

Riyadh 11426; Saudi Arabia

Tel No. +966 (11)429-4444 Ext. 94535

Fax: +966(11)429-4440

Email: matouepnasrisa@mngha.med.sa

Reviewer 2 Report (New Reviewer)

Comments and Suggestions for Authors

In this manuscript, the Authors test the pro-apoptotic properties of alkaloids and show, using KO cell models, that their mechanism of action is via VDR. Although these findings are interesting and novel, the Authors should strengthen data presentation, methodology description and change some statements to fully support the results presented: 

  1. For Figure 1C the Authors state: The VDR appeared highly expressed and was detected within the nucleus and in the cytoplasm, depicting a perinuclear localization (Figure 1C). In HCT116-VDR/KO cells, VDR expression was significantly reduced, confirming the dramatic decrease in VDR expression (Figure 1C).

Is the antibody used for immunofluorescence staining recognizing all 3 VDR isoforms? How did the Authors access reduced expression in this experiment, since all the cells seem to be positive?

  1. Are differences in Figure 1E significant?
  2. In Figure 2 the Authors state: untreated HCT116-VDR/KOcells exhibited minimal green fluorescence in the cytoplasm and nucleus, indicating low overall VDR expression. No noticeable increase in VDR expression or nuclear translocation was observed in any of the treated (VitD3, coclaurine, and reticuline) HCT116-VDR/KO cells, compared with untreated cells.
    There is no evidence for these statements. High expression of VDR is visible in HCT116-VDR/KO cells. The Authors should show enlarged nucleus OF HCT116-VDR/KO cells FOR comparison to WT to convince the readers that there is no nuclear localization. The Authors should quantify VDR HCT116-VDR/KO cells and present them on the graph.
  3. If MTT test was performed to assess the concentrations of coclaurine, reticuline, and VitD3 to be used, this experiment needs to be Figure 3. The MTT test is usually presented as a line chart with points ± error bars to better visualize the decline in viability with increasing concentrations of the substance.
  1. In paragraph 3.6 the Authors state: “As another cell death mechanism, cell cycle distribution was determined…” which is incorrect. Cell cycle can not be a death mechanism and this needs to be rephrased.
  2. On graphs for G0/G1, S, and G2/M the x-axis values need to be standardized, with a recommendation to put them to 100% for better comparison.
  3. Are used CRISPR/Cas9 sgRNA tested for predicted off target effect? This should be stated in the materials and methods.
  4. Authors purchased used alkaloids from a plant extracts from a company. Was purity, solubility and cell permeability of these compounds tested before? The reviewer can not access this info online via catalog number. This should be stated in the Materials and Methods.

Author Response

RESPONSE TO THE REVIEWER #2

Manuscript ID: CIMB-3878374

Title: Reticuline and coclaurine exhibit vitamin D receptor-dependent anticancer and pro-apoptotic activities in colorectal cancer cell lines

Dear Editor,

Thank you for considering this manuscript for review. We are grateful to all reviewers for their detailed feedback and believe that this has improved the quality of our revised manuscript. Please find below our detailed response (in red) to each point raised by the reviewer #2.

______________________________________________________________________________

In this manuscript, the Authors test the pro-apoptotic properties of alkaloids and show, using KO cell models, that their mechanism of action is via VDR. Although these findings are interesting and novel, the Authors should strengthen data presentation, methodology description and change some statements to fully support the results presented: 

  1. For Figure 1C the Authors state: The VDR appeared highly expressed and was detected within the nucleus and in the cytoplasm, depicting a perinuclear localization (Figure 1C). In HCT116-VDR/KO cells, VDR expression was significantly reduced, confirming the dramatic decrease in VDR expression (Figure 1C).

Is the antibody used for immunofluorescence staining recognizing all 3 VDR isoforms? How did the Authors access reduced expression in this experiment, since all the cells seem to be positive?

Thank you for the relevant comment. Regarding immunofluorescence staining, we used the same antibody that was used for Western blotting, the mouse monoclonal anti-VDR antibody [clone D-6] (#sc-13133, Santa Cruz Biotechnology). According to the manufacturer’s information, this monoclonal antibody recognizes an epitope shared among the known human VDR isoforms and is therefore expected to detect all VDR isoforms. To demonstrate the obvious decrease of VDR expression detected in VDR/KO cells compared to the WT cells, we measured the mean of fluorescence intensity using ImageJ sofware and obtained a significant decrease in VDR expression. The analysis represented in bar graph has been added to the revised Figure 1C.

  1. Are differences in Figure 1E significant?

Thank you for the comment. Although differences were clearly observed, there was no significant difference between the cell growth rate assessed between WT and VDR/KO cells.

  1. In Figure 2 the Authors state: untreated HCT116-VDR/KOcells exhibited minimal green fluorescence in the cytoplasm and nucleus, indicating low overall VDR expression. No noticeable increase in VDR expression or nuclear translocation was observed in any of the treated (VitD3, coclaurine, and reticuline) HCT116-VDR/KO cells, compared with untreated cells.
    There is no evidence for these statements. High expression of VDR is visible in HCT116-VDR/KO cells. The Authors should show enlarged nucleus OF HCT116-VDR/KO cells FOR comparison to WT to convince the readers that there is no nuclear localization. The Authors should quantify VDR HCT116-VDR/KO cells and present them on the graph.

Thank you for this comment. We would like to apologize for these figures that eventually became blurry, which made more difficult to see the obvious VDR localizations in WT cells and very low expression in VDR/KO cells. As suggested, we magnified all the high-resolution photomicrographs and enlarged one representative cell for a better observation of VDR expression levels in both cellular compartments (i.e., nucleus and cytoplasm). Statistical analysis was preformed for all conditions (WT and VDR/KO cells) after measurement of the mean of fluorescence intensity as previously described. A new chart is shown in the corresponding revised Figure (new Figure 3).

  1. If MTT test was performed to assess the concentrations of coclaurine, reticuline, and VitD3 to be used, this experiment needs to be Figure 3. The MTT test is usually presented as a line chart with points ± error bars to better visualize the decline in viability with increasing concentrations of the substance.

Thank you for this relevant comment. As suggested, we moved the MTT results just after Figure 1, which confirms the impact of VDR knockdown at the cellular functional level. Thus, MTT results, now presented in line chart, become Figure 2.

  1. In paragraph 3.6 the Authors state: “As another cell death mechanism, cell cycle distribution was determined…” which is incorrect. Cell cycle can not be a death mechanism and this needs to be rephrased.

Thank you for the comment. We totally agree and rephrased it as follows:

“In addition to apoptosis, cell cycle distribution was determined..”.

  1. On graphs for G0/G1, S, and G2/M the x-axis values need to be standardized, with a recommendation to put them to 100% for better comparison.

Thank you for this suggestion. The x-axis of all bar graphs have been standardized to 100%.

  1. Are used CRISPR/Cas9 sgRNA tested for predicted off target effect? This should be stated in the materials and methods.

Thank you for your relevant comment. However, as reported by the company that established the HCT116 VDR-KO cell line (Order SO-10091723-1), they do not experimentally measure the off-target effects of a given sgRNA. They choose guide designs that are predicted to have minimal off-target effects. Moreover, using a SpCas9 protein format rather than plasmid/viral DNA or a SpCas9-expressing cell line restricts SpCas9 expression to a short time window, which can also help reduce off-target editing as described in Cameron et al. (2017, PMID: 28459459), Liang et al. (2015, PMID: 26003884), and Kim et al. (2014, PMID: 24696461).

Therefore, as suggested,  in the Materials and Methods, we revised section 2.2. as follows:

“The design and synthesis of the modified single guide RNA, predicted with minimal off-target effects, and donor template (VDR/KO) were performed by Synthego Corporation CRISPR Genome Engineering Service.”

  1. Authors purchased used alkaloids from a plant extracts from a company. Was purity, solubility and cell permeability of these compounds tested before? The reviewer can not access this info online via catalog number. This should be stated in the Materials and Methods.

Thank you for your comment. Cell-permeability data were not provided by the supplier. However, we have added the requested information on purity and solubility to the Materials and Methods section, under 2.1 Reagents section of the revised manuscript as follows:

“Coclaurine (#SC5960) and reticuline (#SR8320) with purity for high-performance liquid chromatography (HPLC) ≥ 98% were purchased from Solarbio Science and Technology Co., Ltd., Beijing, China).”

Regarding the solubility, we previously mentioned that:

“ Dimethyl sulfoxide (DMSO), the solvent used for the reconstitution of alkaloids..”

We hope these changes are satisfactory as we are keen to publish in Current Issues in Molecular Biology journal. Please feel free to contact us should you require further clarification.

We are looking forward to hearing from you

With my best regards

Sabine Matou-Nasri, PhD.

Senior research scientist

Blood and Cancer Research Department

Affiliate Professor, Biosciences Department, Faculty of the School of Systems Biology, George Mason University, VA, USA

King Abdullah International Medical Research Center (KAIMRC)

Ministry of National Guard-Health Affairs (MNG-HA)

P O Box 22490; Mail Code# 1515

Riyadh 11426; Saudi Arabia

Tel No. +966 (11)429-4444 Ext. 94535

Fax: +966(11)429-4440

Email: matouepnasrisa@mngha.med.sa

Reviewer 3 Report (New Reviewer)

Comments and Suggestions for Authors

In this study, the authors investigated the anticancer and pro-apoptotic effects of coclaurine on colorectal cancer (CRC) cell line. CRISPR/Cas9-edited method was used to establish vitamin D receptor (VDR) knockout cell line (HCT116-KO). Cytotoxicity, apoptosis, cell cycle progression results showed that coclaurine and reticuline exert anti-CRC and pro-apoptotic activities through VDR dependent manner, suggesting them as natural therapeutic candidates. The study focuses on in vitro evaluation, while in vivo investigation is encouraged to be complete. Moreover, there are several issues need to be addressed before the manuscript can be further considerate.

  1. Through the whole manuscript, the authors proved the results only in HCT116 cell line. It is suggested to validate the results on an additional colon cancer line. Otherwise, please revise to “colorectal cancer cell line” in the title.
  2. In Figure 1A, the results showed that VDR isoforms 54 kDa and 48 kDa were significantly downregulated in HCT116-VDR/KO cells. Why did the VDR isoform 60 kDa show no difference between WT and VDR/KO cells? Is this a successful establishment of HCT116-VDR/KO cell line?
  3. In Figure 1C, please include the scale bar in figures. Could the author provide the statistic analysis of VDR fluorescence signal?
  4. In Figure 2A, the authors concluded that “nuclear localization of the VDR was observed in reticuline-treated HCT116-WT cells”. Based on the immunofluorescence results, nuclear localization is not obvious. It is recommended extracting the nuclear and cytoplasmic protein, performing western blot to confirm the nuclear localization of VDR expression.
  5. In Figure 3A, VDR expression upregulated to different extent upon coclaurine and reticuline treatment. Could the author explain why the TP53 level increased to the same extent following these treatments?

Author Response

RESPONSE TO THE REVIEWER #3

Manuscript ID: CIMB-3878374

Title: Reticuline and coclaurine exhibit vitamin D receptor-dependent anticancer and pro-apoptotic activities in colorectal cancer cell lines

Dear Editor,

Thank you for considering this manuscript for review. We are grateful to all reviewers for their detailed feedback and believe that this has improved the quality of our revised manuscript. Please find below our detailed response (in red) to each point raised by the reviewer #3.

______________________________________________________________________________

In this study, the authors investigated the anticancer and pro-apoptotic effects of coclaurine on colorectal cancer (CRC) cell line. CRISPR/Cas9-edited method was used to establish vitamin D receptor (VDR) knockout cell line (HCT116-KO). Cytotoxicity, apoptosis, cell cycle progression results showed that coclaurine and reticuline exert anti-CRC and pro-apoptotic activities through VDR dependent manner, suggesting them as natural therapeutic candidates. The study focuses on in vitro evaluation, while in vivo investigation is encouraged to be complete. Moreover, there are several issues need to be addressed before the manuscript can be further considerate.

  1. Through the whole manuscript, the authors proved the results only in HCT116 cell line. It is suggested to validate the results on an additional colon cancer line. Otherwise, please revise to “colorectal cancer cell line” in the title.

Thank you for this relevant comment. The revised Title is “Reticuline and coclaurine exhibit vitamin D receptor-dependent anticancer and pro-apoptotic activities in the colorectal cancer cell line HCT116”.

  1. In Figure 1A, the results showed that VDR isoforms 54 kDa and 48 kDa were significantly downregulated in HCT116-VDR/KO cells. Why did the VDR isoform 60 kDa show no difference between WT and VDR/KO cells? Is this a successful establishment of HCT116-VDR/KO cell line?

Thank you for this insightful comment. Although the establishment of HCT116-VDR/KO cells using CRISPR-Cas9 did not downregulate the VDR isoform 60 kDa, we were still able to observe a lack of cell response to VitD3 using VDR/KO cells while VitD3 was efficient in VDR-expressing cells (WT). These findings confirm the successful establishment of HCT116-VDR/KO cell line, revealing the key role of VDR isoforms 54kDa and 48 kDa in VDR activities.   

  1. In Figure 1C, please include the scale bar in figures. Could the author provide the statistic analysis of VDR fluorescence signal?

Thank you for your comment. The scale bar has been added in the representative Figure. VDR fluorescence signal was quantified using ImageJ software, generating a bar graph along the representative photomicrographs of Figure 1C.

  1. In Figure 2A, the authors concluded that “nuclear localization of the VDR was observed in reticuline-treated HCT116-WT cells”. Based on the immunofluorescence results, nuclear localization is not obvious. It is recommended extracting the nuclear and cytoplasmic protein, performing western blot to confirm the nuclear localization of VDR expression.

Thank you for your comment and suggestion. After proceeding with the quantification of VDR signal from Figure 1C, same procedure was applied for nuclear and cytoplasmic VDR expression. Thus a new bar graph was generated confirming the significant increase in nuclear and cytoplasmic VDR expression in VitD3, coclaurine and reticuline-treated HCT116-WT cells compared to the untreated cells. Low amount of nuclear and cytoplasmic VDR was detected in HCT116-VDR/KO cells, which was not affected even by the addition of the compounds of interest.

  1. In Figure 3A, VDR expression upregulated to different extent upon coclaurine and reticuline treatment. Could the author explain why the TP53 level increased to the same extent following these treatments?

Thank you for your insightful comment. It is well known that both tumor suppressors VitD and p53 signaling protect against spontaneous or carcinogen-induced malignancy and both promote the transactivation of genes stimulating cell death. VDR is reported as a transciptional target of TP53 and VDR plays a role in p53-mediated suppression of tumor growth, including apoptosis. Thus, we can assume that both alkaloids demonstrated to act through VDR could also positively regulate in the same extent the transcriptional activation of TP53.

We hope these changes are satisfactory as we are keen to publish in Current Issues in Molecular Biology journal. Please feel free to contact us should you require further clarification.

We are looking forward to hearing from you

With my best regards

Sabine Matou-Nasri, PhD.

Senior research scientist

Blood and Cancer Research Department

Affiliate Professor, Biosciences Department, Faculty of the School of Systems Biology, George Mason University, VA, USA

King Abdullah International Medical Research Center (KAIMRC)

Ministry of National Guard-Health Affairs (MNG-HA)

P O Box 22490; Mail Code# 1515

Riyadh 11426; Saudi Arabia

Tel No. +966 (11)429-4444 Ext. 94535

Fax: +966(11)429-4440

Email: matouepnasrisa@mngha.med.sa

Round 2

Reviewer 1 Report (New Reviewer)

Comments and Suggestions for Authors

I would like to thank the authors for addressing the comments and implementing the suggested changes. At this point, I recommend the manuscript for publication in CIMB.

Reviewer 2 Report (New Reviewer)

Comments and Suggestions for Authors

The Authors have adequately addressed all the comments of the reviewer. The addition of fluorescence quantification and enlarged representative images clarified VDR localization. The reorganization of MTT results, as well as rephrasing of cell cycle analysis and standardization of graphs improved data presentation. Inclusion of information on sgRNA off-target effects and alkaloid purity further strengthened Material and Methods section. Together, these revisions have significantly increased the scientific soundness and overall quality of the manuscript.

This manuscript is a resubmission of an earlier submission. The following is a list of the peer review reports and author responses from that submission.

Round 1

Reviewer 1 Report

Comments and Suggestions for Authors

General Summary:

This study investigates the anticancer and pro-apoptotic effects of coclaurine and reticuline, two alkaloids targeting the vitamin D receptor (VDR), using CRISPR/Cas9-edited VDR-knockout (VDR/KO) and wild-type (WT) HCT116 colorectal cancer (CRC) cell lines.While the study addresses an interesting topic, the manuscript requires significant revisions to improve clarity, accuracy, and consistency. Below are detailed comments:

Specific Comments

  1. Please provide the ATCC catalog number for the HCT116 cell line used in this study to ensure reproducibility.
  2. The schematic representation of CRISPR-Cas9 gene editing in Supplementary Figure 1 contains inaccuracies.
  3. Clarification on VDR-KO Cell Generation (CRISPR-Cas9 Methodology):Were the VDR-KO cells obtained from ATCC or generated by authors? The methods section lacks critical details: transfection efficiency of sgRNA and Cas9 protein?
  4. The Materials and Methods section does not provide sufficient detail regarding cell culture and treatment (lines 114-125).
  5. The author describes the cell migration experiment as a scratch wound healing assay. Is this description correct?
  6. Replace "μl" with "μL" (microliter) throughout the entire manuscript.
  7. The value "10,000" is ambiguous—is this cells/well (density) or total cells? Align with the format in Line 158 (e.g., "10,000 cells/well").
  8. The sentence in Lines 194-203 is overly lengthy and unclear. Revise for conciseness and logical flow.
  9. Redundant Terminology (Line 210): "Real-time RT-qPCR" is repetitive. Use either "RT-qPCR" or "quantitative real-time PCR."
  10. Reformulate Table 1 as a three-line table. Name primers (e.g., VDR-F/R) and cite them in the methods.
  11. Image Quality Improvement: The figures, particularly confocal microscopy images, are of low resolution. Provide higher-quality images with clear scale bars and labels.
  12. The content in Lines 506-520 is summative and belongs in the Conclusion section, not the Discussion. Focus the Discussion on mechanistic insights and contextualizing results.
  13. Standardize references: Include DOI and page numbers where missing. Follow journal guidelines.

Author Response

Review 1:
The work is devoted to the promising topic of exosomes and their role in cell-cell
interactions. The author did a great job conducting cellular, biochemical, and bioinformatics
studies. The results are mainly presented clearly and informatively. This is especially
impressive considering the author worked alone. I believe the manuscript deserves to be
published.
However, I have questions about the novelty of the study. It is well-known that tumor and
healthy tissues have different molecular genetic profiles. It is more convenient to examine
the tissues themselves than to examine exosomes. It is also well known that cells can
generate exosomes (broadly defined as conditioned medium) bearing the imprint of their
molecular genetic profile. It is also well known that TGF-beta causes the Smad2/3 and
ERK1/2 cascade; this is the canonical pathway of epithelial-mesenchymal transition (EMT),
which is curiously absent in the manuscript introduction and discussion. Therefore, despite
the large amount of work done, the novelty is limited.
Regarding the novelty of the study:
I appreciate the reviewer’s concern regarding the novelty of our study. The novelty of our
study first lies in our research strategy: We combined single-cell RNA-seq–based
discovery with exosomal transcriptomics and targeted functional validation, thereby
establishing a pipeline that moves from transcriptional inference to mechanistic verification,
reducing unnecessary or non-informative experimentation. This approach not only bridges
bioinformatics with both in vivo and in vitro wet-lab experiments but also minimizes
unnecessary or non-informative testing, ultimately reducing experimental burden and
overall cost.
Specifically, we identified a notable upregulation of exosome-related transcriptional activity
at the single-cell level in AML bone marrow, a feature that has not been extensively
characterized in previous studies. This prompted us to perform bulk transcriptomic profiling
of purified exosomes, revealing elevated translational, transcriptional, and metabolic
signatures in AML-derived exosomes.
Lastly, in addition to finding that AML-derived exosomes are enriched in TGF-β, which
functionally activates the Smad2/3–MMP2 and ERK1/2 signaling pathways, our cell–cell
interaction analysis revealed a significant increase in both the number and strength of
interactions in the AML group compared to healthy controls (Figure 7A). This suggests that
exosome-mediated mechanisms may enhance intercellular communication. Furthermore,
the expression of immune regulatory markers such as FOXP3, CD274, LGALS1, CCL2,
CCL5, IL2RA, LAG3, TNFRSF18, and HAVCR2 were upregulated in the AML group,
indicating activation of immune modulatory processes. These changes may influence T
cell activation, differentiation, migration, and immune checkpoint signaling, thereby
contributing to the remodeling of the immune landscape within the leukemic
microenvironment.
These findings were validated through both in vitro functional assays and in vivo immune
microenvironment modeling, adding translational significance to our observations.
Regarding the TGF-β/Smad/ERK axis:
We appreciate the reviewer’s comments regarding the absence of discussion on the
canonical EMT signaling pathway. To extend the investigation and uncover additional
mechanisms, we performed cell-cell interaction and pathway analyses, which revealed that
AML-derived exosomes significantly upregulate several key immune-related signaling
pathways, including CD99, SELL (L-selectin), MHC class I and II, Semaphorin (SEMA),
NOTCH, and CEACAM. These pathways regulate crucial processes such as antigen
presentation, immune cell trafficking, activation, and immune checkpoint control, and were
predominantly enriched in CD63⁺ exosome-associated populations. However, we have to
admit that more functional wet-lab experiments are required to further validate the specific
contributions of these pathways.
1) Discuss the epithelial-mesenchymal transition in the introduction and results.
A: We thank the reviewer for this valuable suggestion. We have incorporated more of the
relationship between AML, exosomes, TGF-β, and epithelial-mesenchymal transition (EMT)
in the Introduction to provide necessary context. Furthermore, the potential link between
cell-cell interactions and EMT is elaborated upon in the Discussion, highlighting how AML-
derived exosomes may trigger EMT-like functional changes despite the lack of classical
EMT in hematologic malignancies. We discussed data on exosome-mediated signaling
pathways, including TGF-β/Smad and ERK activation, and the upregulation of immune-
related pathways that exhibit parallels to EMT processes. Together, these revisions offer
a comprehensive improvement in EMT relevance to our study.
2) Clearly formulate the novelty, which obviously relates to the exosome part.
A: Our work uniquely demonstrates that AML-derived exosomes exhibit significantly
enhanced transcriptional, translational, and metabolic activities compared to those from
healthy donors. Importantly, we identify that these exosomes are highly enriched in TGF-β
and promote leukemic cell proliferation and migration through activation of the Smad2/3–
MMP2 and ERK1/2 signaling pathways.
Furthermore, we reveal that AML exosomes actively remodel the bone marrow immune
microenvironment by upregulating multiple immunoregulatory pathways, highlighting a
novel mechanism of disease progression. This integrative analysis not only deepens
mechanistic understanding of AML pathogenesis but also highlights exosomes and their
signaling cascades as promising targets for therapeutic intervention.
Accordingly, we have revised the abstract to clearly emphasize these novel findings.
3) Discuss how the exosome profile corresponds to the profile of the original tissue (e.g.,
according to literature data) so that exosomes can be proposed as an object for liquid
biopsy/fingerprinting of tumors.
A: Thank you for your valuable suggestion. In response, we have added more in the
Discussion section highlighting the correspondence between the molecular profiles of
AML-derived exosomes and their cells of origin.
Our integrative analysis reveals a strong correspondence between the molecular profiles
of AML-derived exosomes and their cells of origin, particularly in the advanced stages of
AML. Consistent with literature reports, we observed that exosomal RNA reflects the
transcriptional landscape of leukemic blasts, with significant enrichment in gene programs
related to transcription, translation, and metabolism. Notably, the high levels of TGF-β
found in AML exosomes mirrored the elevated TGF-β expression detected in leukemic cells
using single-cell RNA sequencing, further emphasizing the fidelity of exosomal content to
tumor-specific features.
Functionally, we demonstrated that AML exosomes actively promote cell proliferation in
both in vitro and in vivo models, an effect that aligns with the aggressive proliferative
phenotype commonly seen in advanced AML. This biological activity underscores the
pathophysiological relevance of exosomes and supports their role as active participants in
disease progression.
By integrating transcriptomic, proteomic, and functional assays, we showed that AML
exosomes not only reflect but also recapitulate key molecular and phenotypic
characteristics of the tumor microenvironment. Taken together, these findings suggest that
exosomes can serve as reliable, minimally invasive surrogates for tumor fingerprinting.
Their molecular fidelity and functional relevance make them promising candidates for use
in liquid biopsy-based diagnostics and disease monitoring in AML.
Technical remarks:
4) Some images are difficult to analyze. Please resize them (e.g., Fig. 2B, 5C–E, etc.).
A: We appreciate the reviewer’s feedback regarding the clarity of certain figures. In
response, we have carefully revised and resized the images in Figures 2B and 5C–E to
improve their resolution and readability.
5) On Fig. 4B, the cells look strange. Cell viability is greater than 100%, which is impossible.
If exosomes really enhance cell growth, it should be labeled "cell proliferation" instead of
"cell viability."
A: Thank you for pointing this out. We agree that the label “cell viability” may be misleading
in this context. In Figure 4B, the y-axis reflects relative cell numbers, which indirectly
indicates cell proliferation rather than viability. To avoid confusion, we have revised the
figure label a to indicate “cell proliferation” instead of “cell viability”.
6) The Mann-Whitney U test is recommended for nonparametric data; the Student's t-test
is less appropriate.
A: Thank you for your suggestion regarding statistical analysis. We think that one-way
ANOVA is more appropriate than both the student’s t-test and the Mann-Whitney U test in
our case, as we are comparing multiple groups under a single condition. Accordingly, we
have reanalyzed the relevant data using one-way ANOVA and updated the methods
section in the revised manuscript.

Reviewer 2 Report

Comments and Suggestions for Authors

Dear Editors,

I have reviewed the manuscript entitled, "Reticuline and coclaurine exhibit vitamin D receptor-dependent anticancer and pro-apoptotic activities in colorectal cancer cell lines." While the work presents interesting findings, I believe the authors could significantly strengthen their study by addressing the following:

  1. Ligand Selection: Justification for the selection of reticuline and coclaurine could be improved by incorporating pharmacophore modeling, derived from either target or ligand structures. This would provide a stronger rationale for their choice.

  2. Target Identification: The study could benefit from the inclusion of bioinformatic analyses, such as protein-protein interaction (PPI) studies, to identify other potential biological targets beyond the vitamin D receptor.

  3. Docking Validation: Docking validations are necessary to increase confidence in the molecular docking results.

  4. Control and Data Validation: To enhance the rigor of the study, the authors should include experiments with non-cancerous cells and a known anticancer drug as a positive control for data validation purposes.

Author Response

RESPONSE TO REVIEWER 2
Manuscript: cimb-3779887
Title: Reticuline and coclaurine exhibit vitamin D receptor-dependent anticancer and pro-apoptotic
activities in colorectal cancer cell lines
Dear editor,
Thank you for considering this manuscript for review. We are grateful to all reviewers for their detailed
feedback and believe that this has improved the quality of our revised manuscript. Wherever possible, our
paper has been modified in light of the comments made by the reviewers. Please find below our detailed
response (in red) to each point raised by the reviewers.
Reviewer #2
Dear Editors,
I have reviewed the manuscript entitled, "Reticuline and coclaurine exhibit vitamin D receptor-dependent
anticancer and pro-apoptotic activities in colorectal cancer cell lines." While the work presents interesting
findings, I believe the authors could significantly strengthen their study by addressing the following:
1. Ligand Selection: Justification for the selection of reticuline and coclaurine could be improved
by incorporating pharmacophore modeling, derived from either target or ligand structures. This
would provide a stronger rationale for their choice.
Thank you for this valuable suggestion. We would like to clarify that the selection of reticuline and
coclaurine was based on a previous in silico study “In Silico Molecular Docking Analysis of the
Potential role of Reticuline and Coclaurine as Anti-colorectal Cancer Alkaloids” (Alghamdi and
Al-Zahrani, J. Pharm. Res. Int., 2022), in which both compounds showed strong binding affinity
to several colorectal cancer targets. This provided a solid rationale to investigate their VDR-
mediated anticancer activity. Since our study focuses on these two specific compounds and not on
a large chemical library, pharmacophore modeling was not applicable.
2. Target Identification: The study could benefit from the inclusion of bioinformatic analyses,
such as protein-protein interaction (PPI) studies, to identify other potential biological targets
beyond the vitamin D receptor.
We appreciate your recommendation regarding additional target identification. In this study, our
primary objective was to investigate the interaction of the selected compounds with the vitamin D
receptor due to its central role in relevant biological pathways and its previously reported
association with these compounds. While PPI or other bioinformatic analyses could reveal
additional targets, this was beyond the scope of the current work. We agree that this approach could
provide valuable insights and will consider integrating such analyses in future studies.
3. Docking Validation: Docking validations are necessary to increase confidence in the molecular
docking results.
Thank you for highlighting this important aspect. We performed validation using a one-step
docking protocol, where the native ligand of the vitamin D receptor was re-docked into its binding
site. This process ensured that the obtained docking results were consistent with the original crystal
structure in terms of binding score and key molecular interactions. These steps confirm the
reliability and reproducibility of our docking methodology.
4. Control and Data Validation: To enhance the rigor of the study, the authors should include
experiments with non-cancerous cells and a known anticancer drug as a positive control for data
validation purposes.
We thank the reviewer for this important suggestion. The primary objective of our current study
was to investigate the differential effects of reticuline and coclaurine in HCT116 colorectal cancer
cells with and without VDR knockout, to specifically explore VDR-dependent anticancer
mechanisms.
While non-cancerous cell lines and a known anticancer drug were not included in this initial
mechanistic study, we agree that such controls are important for full biological validation. These
experiments are planned as part of our next phase of in vitro and in vivo studies, where we will
assess the selectivity, cytotoxicity, and comparative efficacy of these alkaloids in a broader
biological context.
We hope these changes are satisfactory. Please feel free to contact us should you require further
clarification.
We are looking forward to hearing from you
With my best regards
Dr Sabine Matou-Nasri
Senior research scientist
Blood and Cancer Research Department
King Abdullah International Medical Research Center (KAIMRC)
King Saud bin Abdulaziz University for Health Sciences
Ministry of National Guard – Health Affairs
P.O. Box 22490, Riyadh 11426
Kingdom of Saudi Arabia
Tel. No.: +966 (11) 429 4535
Fax No.: +966 (11) 429 4440
Email: matouepnasrisa@mngha.med.sa

Reviewer 3 Report

Comments and Suggestions for Authors

Dear Author, I give you the following comment. Please address this in your manuscript to enhance the readability and understanding of your manuscript.

Major Comments :

  1. How does the study clearly differentiate the novelty of coclaurine and reticuline’s mechanism from previously reported VDR ligands or anticancer agents?
  2. Why were only in vitro models (HCT116-WT and HCT116-VDR/KO) used, and how might this limit the translatability of the findings?
  3. What specific rationale guided the selection of coclaurine and reticuline among the broad class of alkaloids?
  4. Could the authors elaborate on whether the observed S-phase arrest is a direct consequence of VDR activation or an indirect downstream effect?
  5. Have the authors considered potential off-target effects of the alkaloids, especially in VDR/KO cells where effects might still be partially observed?

Minor Comments :

  1. Could the authors include a schematic figure in the Introduction to visually highlight the novelty and mechanism of their study?
  2. Have the authors ensured that all abbreviations (e.g., PARP, Bax, TP53) are clearly defined upon first use in the manuscript?
  3. Are the results of the molecular docking quantitatively summarized (binding energy, RMSD values) for better interpretation?
  4. Is the scratch assay time and quantification method clearly described in the methods section?
  5. Are there any data supporting the bioavailability or pharmacokinetic properties of coclaurine and reticuline in vivo, which could be briefly discussed?

These questions aim to address both overarching concerns and specific technical details that could impact the robustness and clarity of the study's findings.

Best Regards,

Author Response

RESPONSE TO REVIEWER 3
Manuscript: cimb-3779887
Title: Reticuline and coclaurine exhibit vitamin D receptor-dependent anticancer and pro-apoptotic
activities in colorectal cancer cell lines
Dear editor,
Thank you for considering this manuscript for review. We are grateful to all reviewers for their detailed
feedback and believe that this has improved the quality of our revised manuscript. Wherever possible, our
paper has been modified in light of the comments made by the reviewers. Please find below our detailed
response (in red) to each point raised by the reviewers.
Reviewer #3
Dear Author, I give you the following comment. Please address this in your manuscript to enhance the
readability and understanding of your manuscript.
Major Comments:
1. How does the study clearly differentiate the novelty of coclaurine and reticuline’s mechanism from
previously reported VDR ligands or anticancer agents?
We appreciate your insightful question. To address this point, we have highlighted the originality
of our findings in the discussion section. Our study shows that coclaurine and reticuline have pro-
apoptotic and anti-migratory effects VDR-dependent. These alkaloids activated intrinsic apoptotic
markers (cleaved PARP and Bax), caused S-phase arrest, and upregulated TP53, indicating a
mechanism different from that of traditional VDR ligands. Additionally, molecular docking
demonstrated that coclaurine and reticuline had binding affinities to the VDR ligand-binding
domain, confirming their direct interactions. These results point to a distinct VDR-modulated
signaling pathway that has not been described for this class of naturally occurring alkaloids before.
2. Why were only in vitro models (HCT116-WT and HCT116-VDR/KO) used, and how might this limit
the translatability of the findings?
We value your perceptive question The deliberate initial step in investigating the VDR-dependent
processes underlying the actions of coclaurine and reticuline involved the utilization of in vitro
models, notably the HCT116-WT and HCT116-VDR/KO cell lines. These isogenic models
allowed us to directly assess the contribution of VDR expression to the observed cellular responses,
including cell cycle arrest, migration inhibition, and apoptosis. We acknowledge that the clinical
applicability of the findings may be constrained by its reliance on in vitro systems. The complexity
of the tumor microenvironment, its interactions with the immune system, and pharmacokinetic
properties cannot be sufficiently modeled in vitro; hence, in vivo models are essential. In the
discussion section we mention the need for more study using animal models to confirm the
therapeutic potential, bioavailability, and safety profile of coclaurine and reticuline in colorectal
cancer.
3. What specific rationale guided the selection of coclaurine and reticuline among the broad class of
alkaloids?
We appreciate your relevant question. The selection of coclaurine and reticuline was based on
several significant considerations. First, the medicinal plant Annona muricata, which has long been
used to treat inflammation and cancer, naturally contains both alkaloids known as
benzylisoquinoline. Although their mechanisms of action are largely unknown, preliminary
screening of phytochemicals and the literature currently show that these compounds have cytotoxic
effects on a number of cancer cell lines.
Second, in silico molecular docking studies revealed that coclaurine and reticuline had favorable
binding affinities to the VDR ligand-binding domain, suggesting that they could be new
medications that interact with VDR. From a scientific standpoint, these alkaloids were a suitable
choice supported by computational prediction and the fact that they had never been investigated in
the context of VDR-modulated colorectal cancer.
4. Could the authors elaborate on whether the observed S-phase arrest is a direct consequence of VDR
activation or an indirect downstream effect?
Thank you for your question. According to our data, reticuline and coclaurine primarily caused the
cell growth arrest in S-phase in HCT116-WT cells but not in VDR/KO cells. Although this implies
that the action is VDR-dependent, the precise molecular mechanism underlying it is still unknown.
It is still uncertain whether S-phase arrest is a direct result of VDR activation or an indirect
downstream effect made possible by VDR-regulated genes, such as TP53 or cell cycle-related
cyclins. Given the complexity of VDR signaling, downstream transcriptional control rather than
direct interaction with cell cycle machinery may be the cause of the arrest.
To clarify the direct and indirect roles of VDR in regulating cell cycle progression and S-phase
arrest, more studies employing transcriptome profiling and chromatin immunoprecipitation
targeting cyclins could be required.
5. Have the authors considered potential off-target effects of the alkaloids, especially in VDR/KO cells
where effects might still be partially observed?
We appreciate your insightful observation. While our findings demonstrate the anti-proliferative
and pro-apoptotic effects of coclaurine and reticuline in HCT116-WT cells compared to HCT116-
VDR/KO cells, we are aware that these alkaloids can interact with different cellular targets, and
the response that was observed in VDR/KO cells may be the result of non-specific lethal effects or
alternative signaling pathways. In the discussion section we mentioned the potential off-target
activities and how crucial it is for future research to identify additional potential molecular targets
and validate specificity through more extensive genetic or pharmacological techniques.
Minor Comments :
1. Could the authors include a schematic figure in the Introduction to visually highlight the novelty
and mechanism of their study?
Thank you for your insightful suggestion. A graphical abstract summarizing the novelty of the
study has been submitted.
2. Have the authors ensured that all abbreviations (e.g., PARP, Bax, TP53) are clearly defined upon
first use in the manuscript?
We appreciate your concern. We have gone over the whole document carefully to make sure that
all the abbreviations, such as PARP, Bax, TP53, and Bcl-2, are clearly defined at the first time they
are used. At the end of the document, there is also a full list of abbreviations.
3. Are the results of the molecular docking quantitatively summarized (binding energy, RMSD
values) for better interpretation?
Thank you for your feedback regarding binding energy and RMSD. In this work, we focused on
molecular docking, which provides a docking score and predicted binding modes rather than the
actual binding free energy. Calculating accurate binding energy and RMSD usually involves
molecular dynamics simulations, which are beyond the current scope but can be considered in
future stages.
4. Is the scratch assay time and quantification method cslearly described in the methods section?
Thank you for this comment. The scratch assay time and quantification method are described in the
methods section (Section 2.6). Photographs of the scratch were taken at 0 and 96 h without
treatment and after 48 h of treatment. The wound closure was quantified by measuring three
different regions in three different images for each condition with ImageJ software version 1.53e.
5. Are there any data supporting the bioavailability or pharmacokinetic properties of coclaurine and
reticuline in vivo, which could be briefly discussed?
Thank you for your thoughtful suggestion. The bioavailability or pharmacokinetic properties of
coclaurine and reticuline are limited. Therefore, further in vivo studies are needed to characterize
the pharmacokinetics, tissue distribution, and safety of these alkaloids as potential clinical
applications.
We hope these changes are satisfactory. Please feel free to contact us should you require further
clarification.
We are looking forward to hearing from you
With my best regards
Dr Sabine Matou-Nasri
Senior research scientist
Blood and Cancer Research Department
King Abdullah International Medical Research Center (KAIMRC)
King Saud bin Abdulaziz University for Health Sciences
Ministry of National Guard – Health Affairs
P.O. Box 22490, Riyadh 11426
Kingdom of Saudi Arabia
Tel. No.: +966 (11) 429 4535
Fax No.: +966 (11) 429 4440
Email: matouepnasrisa@mngha.med.sa

Reviewer 4 Report

Comments and Suggestions for Authors
  1. Alkaloids are of particular interest as potential anticancer agents. – Sentence need to be checked?
  2. Vitamin D receptor (VDR) plays a role in preventing colorectal cancer (CRC) and may be crucial for alkaloids anticancer properties. – Sentence needs to be check and rewrite.
  3. Objective of the study not clear in abstract, it should be modify.
  4. Numerous phytochemicals, particularly alkaloids, extracted from A. muricata have garnered a particular attention for their anti-inflammatory and anticancer properties against the liver, lung, prostate, pancreas, breast and colon cancers. – Alkaloids extracted from which part of A. muricata. It should be included in the manuscript.
  5. In introduction, background of coclaurine and reticuline needs to be added in the manuscript.
  6. In MTT methodology, what are the concentration was used for your study. It should be included in the manuscript.
  7. Line 171 - Briefly, the cells were collected – How it was collected? It should be included in the manuscript.
  8. In methodology, MTT should be reported first, because it was the concentration confirmatory study. So it should be modify.
  9. All the experimental methodology, treatment concentration should be included.
  10. In the results, why are you reported western blotting first? It was stated in the end of methodology. All the results should be modify based on your methodology.
  11. Line 266 - As shown in representative photomicrographs, after 96 h of incubation, the gap size in the wounded HCT116-WT cell monolayer was larger than that measured in the HCT116-VDR/KO cell monolayer, demonstrating that HCT116-VDR/KO cells exhibited significantly faster wound healing than HCT116-WT cells. But you are stated in your methodology the final treatment hours was 72, can you please explain this? It was included in the manuscript.
  12. Loss of VDR expression is well known to promote tumor development and progression [27,28]. – Move to discussion part.
  13. When activated by its ligand, VDR exhibits a perinuclear localization, is then translocated into the nucleus and subsequently binds to target genes for expression, making its cellular localization an indicator of its activity [29]. – What ligand? This part move to discussion.
  14. As shown in Figure 4, after 48 h of incubation, VitD3, coclaurine, and reticuline significantly reduced the wound healing process capacity of HCT116-WT cells compared to control (untreated) and vehicle control (DMSO-treated) cells. – Here my question is, as per your results wound healing process capacity was reduced in HCT116-WT means it was increased the wound? Can you please explain this?
  15. As per your wound measurement graph was showed, HCT116-WT was reduced which means wound healing capacity was increased. Can you please explain this?
  16. Therefore, as another cell death mechanism, cell cycle progression was analyzed in HCT116-WT and HCT116-VDR/KO cells exposed to VitD3, coclaurine, and reticuline, followed by DNA staining with PI and monitoring of DNA content in percentage of cells using flow cytometry software. – Flow cytometry software or test? Software means include the specific software name in the manuscript.
  17. Using Western blot analysis, a significant increase in cyclins A1 and B1 expression level was observed in coclaurine and reticuline-treated HCT116-WT cells compared to DMSO-treated cells (Figure 7C). – What about control cells (without treatment)?
  18. The use of VDR/KO CRC cell line revealed the crucial role of VDR in coclaurine and reticuline stimulatory effects on the upregulation of VDR and tumor suppressor TP53, and on their inhibitory effects on wound healing process and cell viability, compared to WT-HCT116 cells. – Wound healing process inhibit means, wound was increased right? What was your conclusion here?
  19. Line 584 - Peganum harmala – Plant name should be in italics.
  20. Demonstrated to act through VDR, the pro-apoptotic effects of coclaurine and reticuline require in-depth investigation focusing on other proteins leading to cell death, such as cyclins causing cell growth arrest – What demonstrated?

Author Response

RESPONSE TO REVIEWER 4
Manuscript: cimb-3779887
Title: Reticuline and coclaurine exhibit vitamin D receptor-dependent anticancer and pro-apoptotic
activities in colorectal cancer cell lines
Dear editor,
Thank you for considering this manuscript for review. We are grateful to all reviewers for their detailed
feedback and believe that this has improved the quality of our revised manuscript. Wherever possible, our
paper has been modified in light of the comments made by the reviewers. Please find below our detailed
response (in red) to each point raised by the reviewers.
Reviewer #3
1. Alkaloids are of particular interest as potential anticancer agents. – Sentence need to be checked?
Thank you for your concern. The sentence has been revised and edited as follows:
“Alkaloids have garnered significant interest as potential anticancer agents.”
2. Vitamin D receptor (VDR) plays a role in preventing colorectal cancer (CRC) and may be crucial
for alkaloids anticancer properties. – Sentence needs to be check and rewrite.
Once again, thank you for your concern. The sentence has been revised and edited as follows:
“The vitamin D receptor (VDR) plays a role in preventing the progression of colorectal cancer (CRC)
and may be a crucial mediator of the anticancer effects produced by certain alkaloids.”
3. Objective of the study not clear in abstract, it should be modify.
Thank you for your relevant comment. Our main objective has been clarified by including this
sentence:
“The search for novel anticancer drug that induce VDR expression and act through the VDR could
improve the clinical outcome of CRC patients.”
4. Numerous phytochemicals, particularly alkaloids, extracted from A. muricata have garnered a
particular attention for their anti-inflammatory and anticancer properties against the liver, lung,
prostate, pancreas, breast and colon cancers. – Alkaloids extracted from which part of A. muricata.
It should be included in the manuscript.
Thank you for your comment. However, we would like to clarify that this information was already
included in the manuscript, In the introduction section, as follows:
“A. muricata produces seven isoquinoline alkaloids, including reticuline and coclaurine, primarily
in its leaves, roots, and stem barks [18]”.
5. In introduction, background of coclaurine and reticuline needs to be added in the manuscript.
As requested, the background of coclaurine and reticuline has been added as follows:
“Reticuline is the important branch point in the biosynthesis of most benzylisoquinoline alkaloids
[19]. Coclaurine has demonstrated anticancer activities against human colorectal cancer (HCT116)
and breast cancer (MCF-7) cell lines in vitro [20].”
6. In MTT methodology, what are the concentration was used for your study. It should be included in
the manuscript.
The concentrations have been added to the manuscript, various concentrations (0.001–20 M) of
VitD3, coclaurine, and reticuline.
7. Line 171 - Briefly, the cells were collected – How it was collected? It should be included in the
manuscript.
The method of cell collection has been added to the manuscript. Briefly, the cells were trypsinized,
collected via centrifugation at 300 × g for 5 min.
8. In methodology, MTT should be reported first, because it was the concentration confirmatory
study. So it should be modify.
About the order of the methodology part. We first have to present the experiments that were done
to confirm the biological impact of the knockout of VDR (such as immunofluorescence staining,
wound healing assay, growth rate, Western blotting, and RT-qPCR). These experiments were
necessary to confirm the knockout on VDR expression level and CRC cellular functions before
moving on to the treatment and experiments. After properly validating the VDR-KO and WT cells,
the MTT assay was done as a step to define the effective concentrations.
9. All the experimental methodology, treatment concentration should be included.
We have revised the Materials and Methods section to make sure that all treatment concentrations
used in the experiments are clearly stated.
10. In the results, why are you reported western blotting first? It was stated in the end of methodology.
All the results should be modify based on your methodology.
Thank you for your relevant comment. However, as mentioned before (response 8), Western blot
was one of the technical assays validating the knockout of VDR, which must be presented early in
the result section along with other validation experiments (such as immunofluorescence staining,
wound healing assay, growth rate, and RT-qPCR), to set up the cell model before proceeding to the
analysis of the therapy.
11. Line 266 - As shown in representative photomicrographs, after 96 h of incubation, the gap size in
the wounded HCT116-WT cell monolayer was larger than that measured in the HCT116-VDR/KO
cell monolayer, demonstrating that HCT116-VDR/KO cells exhibited significantly faster wound
healing than HCT116-WT cells. But you are stated in your methodology the final treatment hours
was 72, can you please explain this? It was included in the manuscript.
That is correct and we can clarify these different incubation time periods. Photomicrographs in the
wound healing assay were taken after 96 h of incubation using the untreated HCT116-WT and
HCT116-VDR/KO cells (Figure 1D) to compare the migration between each cell line, and after 48
h with the cells upon treatment (Figure 4). The 72 h refers to another assay, such as MTT cell
viability, but not for wound healing assay.
12. Loss of VDR expression is well known to promote tumor development and progression [27,28]. –
Move to discussion part.
Thank you for your relevant suggestion. This has been moved to the Discussion section.
13. When activated by its ligand, VDR exhibits a perinuclear localization, is then translocated into the
nucleus and subsequently binds to target genes for expression, making its cellular localization an
indicator of its activity [29]. – What ligand? This part move to discussion.
The ligand for the (VDR) is 1,25-dihydroxyvitamin D₃ (VitD₃) and the cited part has also been
moved to the Discussion as follows:
“The VDR, a ligand-activated transcription factor and a relevant prognostic marker for CRC
patients, forms a heterocomplex with its main ligand, VitD3, which subsequently prevents CRC
and even inhibits CRC development and progression [7, 33]. When activated by its ligand, the VDR
exhibits a perinuclear localization, is then translocated into the nucleus, and subsequently binds
to target genes for expression, making its cellular localization an indicator of its activity [35].”
14. As shown in Figure 4, after 48 h of incubation, VitD3, coclaurine, and reticuline significantly
reduced the wound healing process capacity of HCT116-WT cells compared to control (untreated)
and vehicle control (DMSO-treated) cells. – Here my question is, as per your results wound healing
process capacity was reduced in HCT116-WT means it was increased the wound? Can you please
explain this?
That is correct. Lower wound healing capacity means that the treated HCT116-WT cells migrated
more slowly, which led to less scratch gap closure and a larger residual wound area after 48 h than
the controls.
15. As per your wound measurement graph was showed, HCT116-WT was reduced which means
wound healing capacity was increased. Can you please explain this?
A lower value in the bar graph indicates a larger remaining wound gap, demonstrating a lower
capacity for wound healing. Therefore, the decreased bar height in HCT116-WT groups treated
indicates an inhibition in wound closure and cell migration.
16. Therefore, as another cell death mechanism, cell cycle progression was analyzed in HCT116-WT
and HCT116-VDR/KO cells exposed to VitD3, coclaurine, and reticuline, followed by DNA
staining with PI and monitoring of DNA content in percentage of cells using flow cytometry
software. – Flow cytometry software or test? Software means include the specific software name
in the manuscript.
For cell cycle analysis, we used ModFit LT™ software (Verity Software House, Topsham, ME,
USA). The name of the software has been included in the manuscript.
17. Using Western blot analysis, a significant increase in cyclins A1 and B1 expression level was
observed in coclaurine and reticuline-treated HCT116-WT cells compared to DMSO-treated cells
(Figure 7C). – What about control cells (without treatment)?
In our experiments, DMSO was used as the vehicle control, since all of the compounds were soluble
and reconstituted in DMSO. As a result, rather than using untreated cells, comparisons were made
with the DMSO group. Although the untreated control group was included in the initial
optimization, it was excluded from the final immunoblot analysis because no significant difference
was observed between the untreated and DMSO-treated samples.
18. The use of VDR/KO CRC cell line revealed the crucial role of VDR in coclaurine and reticuline
stimulatory effects on the upregulation of VDR and tumor suppressor TP53, and on their inhibitory
effects on wound healing process and cell viability, compared to WT-HCT116 cells. – Wound
healing process inhibit means, wound was increased right? What was your conclusion here?
Correct. Wound healing inhibition → the wound gap remains larger → less cell migration/closure.
Our conclusion: these results demonstrate that reticuline and coclaurine have anti-migratory effects
on VDR-expressing CRC. Both alkaloids increased TP53 and VDR expression in HCT116-WT
cells, which decreased cell migration (impaired wound healing). However, in HCT116-VDR/KO
cells, these effects were not observed, indicating that VDR plays a crucial role in mediating these
alkaloids' anticancer effects.
19. Line 584 - Peganum harmala – Plant name should be in italics.
Thank you for your relevant comment. The plant name is now in italics.
20. Demonstrated to act through VDR, the pro-apoptotic effects of coclaurine and reticuline require in-
depth investigation focusing on other proteins leading to cell death, such as cyclins causing cell
growth arrest – What demonstrated?
Thank you for the insightful comment. The anti-proliferative effects of alkaloids were demonstrated
to act through VDR. However, to avoid any confusion and for clarification, the sentence has been
changed as follows:
“The VDR-dependent pro-apoptotic effects of coclaurine and reticuline require in-depth
investigation focusing on other proteins leading to cell death, such as cyclins causing cell growth
arrest.”
We hope these changes are satisfactory. Please feel free to contact us should you require further
clarification.
We are looking forward to hearing from you
With my best regards
Dr Sabine Matou-Nasri
Senior research scientist
Blood and Cancer Research Department
King Abdullah International Medical Research Center (KAIMRC)
King Saud bin Abdulaziz University for Health Sciences
Ministry of National Guard – Health Affairs
P.O. Box 22490, Riyadh 11426
Kingdom of Saudi Arabia
Tel. No.: +966 (11) 429 4535
Fax No.: +966 (11) 429 4440
Email: matouepnasrisa@mngha.med.sa

Reviewer 5 Report

Comments and Suggestions for Authors

The manuscript "Reticuline and coclaurine exhibit vitamin D receptor-dependent anticancer and pro-apoptotic activities in colorectal cancer cell lines" submitted for review is devoted to a rather interesting and promising problem of searching for natural anticancer agents. The text is written quite сoncisely and clearly, the design of the experiments is logical, and most of the conclusions are well-founded. The discussion creates a sufficient context for the results. 

Nevertheless, during the review, I found a number of shortcomings that require the attention of the authors.

  1. Omitted symbols in lines 116 and 320.
  2. The term "vehicle" in line 125 et seq. is not widely used in the context used. It probably should be changed.
  3. The authors should add a reference from which the мethodology for the Scratch wound healing assay was taken. It is also unclear from the description provided whether the medium that was injected into the wells after washing contained fetal serum or not. There is an extra symbol in line 155.
  4. The authors indicated that they used the Cell Meter Colorimetric MTT assay (#M8180) (line 160) to perform the MTT test. However, there is no item with that name and catalog number in the catalog of the Cell Meter brand. This section needs to be edited, the authors need to specify the actual name and source of the reagents used. This is necessary so that it is clear which manufacturer's instructions were used, as well as that the results of the MTT test were reproducible. The methodology in the section is also described very briefly, omitting many important parts for reproducibility (for example, what was the volume of the medium in the wells, whether the medium was aspirated before adding DMSO, etc.). In its current form, section 2.7 does not allow reproducing the results of the authors of the manuscript.
  5. Table 1 must be reformatted, it cannot be published in its current form.
  6. The resolution of the illustrations is very low. All figures should be presented in much better quality. In its current form, the text on them is difficult to read, especially on the axes. In addition, the figures are strongly shifted to the right.
  7. From the legend of Figure 1D, it is unclear which specific treatment variants were used for the photos shown. Additionally, in lines 267-270 of the text, the authors state "the gap size in the wounded HCT116-WT cell monolayer was larger than that measured in the HCT116-VDR/KO cell monolayer, demonstrating that HCT116-VDR/KO cells exhibited significantly faster wound healing than HCT116-WT cells". However, it is not clear whether this conclusion was based on a statistical analysis or not, as the figure does not provide any evidence for this.
  8. It is also unclear how well-founded and reliable the authors' conclusion is in lines 274-275, where they state "CT116-VDR/KO cells showed a more pronounced increase in cell growth than HCT116-WT cells". Figure 1E does not contain a data that could be the basis for such a conclusion.
  9. In the legend of Figure 1, there are designations for "*", "***" and "****", however, the figure itself lacks "****" and contains "**", the designation for which is not presented.
  10. The data presented by the authors in Figure 2 does not allow us to confirm or refute their conclusions in lines 303-315. If desired, you can find structures on any of the presented photographs, which the authors described as "an example of a VDR nuclear location." It is necessary to analyze the numerical data of the normalized values of the intensity of green fluorescence in the cytoplasm and the nuclear region so that similar conclusions can be drawn.
  11. In the legend of Figure 4, the authors should specify the incubation time.
  12. The dotplots shown in Figure 6A look sloppy. Apparently, the authors did not use sufficient compensation when forming the protocol. The FACS Diva software allows you to apply compensation after receiving the results. This must be done because insufficient compensation can lead to errors in the analysis of the distribution between subpopulations of early/late apoptotic cells.
  13. It may also help to analyze the fluorescence of PI in the PerCP channel, where the need for compensation is much less pronounced. This should at least be attempted, because although the general trend detected by the authors is not in doubt, there may be errors in determining the proportions of subpopulations.

Author Response

RESPONSE TO REVIEWER 5
Manuscript: cimb-3779887
Title: Reticuline and coclaurine exhibit vitamin D receptor-dependent anticancer and pro-apoptotic
activities in colorectal cancer cell lines
Dear editor,
Thank you for considering this manuscript for review. We are grateful to all reviewers for their detailed
feedback and believe that this has improved the quality of our revised manuscript. Wherever possible, our
paper has been modified in light of the comments made by the reviewers. Please find below our detailed
response (in red) to each point raised by the reviewers.
Reviewer #4
The manuscript "Reticuline and coclaurine exhibit vitamin D receptor-dependent anticancer and pro-
apoptotic activities in colorectal cancer cell lines" submitted for review is devoted to a rather interesting
and promising problem of searching for natural anticancer agents. The text is written quite сoncisely and
clearly, the design of the experiments is logical, and most of the conclusions are well-founded. The
discussion creates a sufficient context for the results.
Nevertheless, during the review, I found a number of shortcomings that require the attention of the authors.
1. Omitted symbols in lines 116 and 320.
Thank you for your comment. These symbols have been edited.
2. The term "vehicle" in line 125 et seq. is not widely used in the context used. It probably should be
changed.
Agree. DMSO was therefore mentioned to be used as the solvent to dissolve the test compounds
and served as the negative control.
3. The authors should add a reference from which the мethodology for the Scratch wound healing
assay was taken. It is also unclear from the description provided whether the medium that was
injected into the wells after washing contained fetal serum or not. There is an extra symbol in line
155.
Thank you for your relevant comment. The methodology was taken from: Al-Nasser et al (2024)
A benzimidazole-based N-heterocyclic carbene derivative exhibits potent antiproliferative and
apoptotic effects against colorectal cancer. Medicina (Kaunas) 60: 1379, and has been added as
reference 24. The medium used was complete medium containing 10% heat-inactivated fetal
bovine serum, 100 IU/ml penicillin, 100 μg/ml streptomycin, and 2 mM L-glutamine. The symbol
in line 155 has been corrected: 4x magnification was edited to 4×magnification.
4. The authors indicated that they used the Cell Meter Colorimetric MTT assay (#M8180) (line 160)
to perform the MTT test. However, there is no item with that name and catalog number in the
catalog of the Cell Meter brand. This section needs to be edited, the authors need to specify the
actual name and source of the reagents used. This is necessary so that it is clear which
manufacturer's instructions were used, as well as that the results of the MTT test were reproducible.
The methodology in the section is also described very briefly, omitting many important parts for
reproducibility (for example, what was the volume of the medium in the wells, whether the medium
was aspirated before adding DMSO, etc.). In its current form, section 2.7 does not allow
reproducing the results of the authors of the manuscript.
We apologize for any confusion. The assay used was MTT Cell Proliferation Assay Kit (#M8180)
from Solarbio Life Sciences (Beijing, China), not Cell Meter, which is now modified in the revised
manuscript. Furthermore, we updated Section 2.7 to include more methodological details.
5. Table 1 must be reformatted, it cannot be published in its current form.
Agree. As requested, Table 1 has been reformatted in the revised manuscript.
6. The resolution of the illustrations is very low. All figures should be presented in much better
quality. In its current form, the text on them is difficult to read, especially on the axes. In addition,
the figures are strongly shifted to the right.
Totally agree. The resolution of each Figure has been improved and centered as recommended.
7. From the legend of Figure 1D, it is unclear which specific treatment variants were used for the
photos shown. Additionally, in lines 267-270 of the text, the authors state "the gap size in the
wounded HCT116-WT cell monolayer was larger than that measured in the HCT116-VDR/KO cell
monolayer, demonstrating that HCT116-VDR/KO cells exhibited significantly faster wound
healing than HCT116-WT cells". However, it is not clear whether this conclusion was based on a
statistical analysis or not, as the figure does not provide any evidence for this.
Thank you for your relevant comment and we apologize for this confusion. In Figure 1D, wound
healing assay comparing the basal migratory ability of HCT116-WT and HCT116-VDR/KO cells.
Cells were cultured without treatment, and scratch wounds were introduced at 0 h. Representative
images of the wound area were taken at 0 and 96 h. Quantification of wound closure was performed
using ImageJ software. Each experiment was repeated three independent times and presented as
mean ± SD. Statistical significance was determined by unpaired Student t-test (*p < 0.05, ***p <
0.001).
8. It is also unclear how well-founded and reliable the authors' conclusion is in lines 274-275, where
they state "CT116-VDR/KO cells showed a more pronounced increase in cell growth than
HCT116-WT cells". Figure 1E does not contain a data that could be the basis for such a conclusion.
We appreciate your insightful comment. We have revised the statement in lines 274–275 to:
However, after 48 hours, HCT116-VDR/KO cells exhibited a slight increase in cell growth
compared to HCT116-WT cells, showing that increased cell growth may be linked to low VDR
expression.
9. In the legend of Figure 1, there are designations for "*", "***" and "****", however, the figure
itself lacks "****" and contains "**", the designation for which is not presented.
Thank you for paying attention to these mistakes. The legend has been corrected.
10. The data presented by the authors in Figure 2 does not allow us to confirm or refute their
conclusions in lines 303-315. If desired, you can find structures on any of the presented
photographs, which the authors described as "an example of a VDR nuclear location." It is
necessary to analyze the numerical data of the normalized values of the intensity of green
fluorescence in the cytoplasm and the nuclear region so that similar conclusions can be drawn.
Thank you for your relevant comment. We prioritized the quantification of the nuclear VDR
localization, the active form, over the predominant cytoplasmic VDR localization, which is also
observed in untreated cells. Therefore, we added the bar graph showing the quantification of
cytoplasmic and nuclear VDR at all conditions. The quantification was proceeded as previously
described in:
El-Obeid A, Maashi Y, AlRoshody R, Alatar G, Aljudayi M, Al-Eidi H, AlGaith N, Khan AH,
Hassib A, Matou-Nasri S. Herbal melanin modulated PGE2 and IL-6 gastroprotective markers
through COX-2 and TLR4 signaling in the gastric cancer cell lines AGS. BMC Complement Med
Ther 2023;23:305.
11. In the legend of Figure 4, the authors should specify the incubation time.
Thank you for your comment. The incubation time has been specified in the legend of Figure 4,
as requested.
12. The dotplots shown in Figure 6A look sloppy. Apparently, the authors did not use sufficient
compensation when forming the protocol. The FACS Diva software allows you to apply
compensation after receiving the results. This must be done because insufficient compensation can
lead to errors in the analysis of the distribution between subpopulations of early/late apoptotic cells.
Thank you for your relevant suggestion, which will be applied in our next in vitro investigation on
apoptosis using flow cytometry, in order to improve the FACS analysis that has been applied and
published in most of our apoptosis-related studies.
13. It may also help to analyze the fluorescence of PI in the PerCP channel, where the need for
compensation is much less pronounced. This should at least be attempted, because although the
general trend detected by the authors is not in doubt, there may be errors in determining the
proportions of subpopulations.
We appreciate the suggestion regarding using the PerCP channel in the detection of PI fluorescence.
In our current study, we used the PE (or FL2) channel for PI fluorescence that aligns with our flow
cytometry setup. We will take your recommendation into consideration in our future work to
improve the accuracy of subpopulation discrimination.
We hope these changes are satisfactory. Please feel free to contact us should you require further
clarification.
We are looking forward to hearing from you
With my best regards
Dr Sabine Matou-Nasri
Senior research scientist
Blood and Cancer Research Department
King Abdullah International Medical Research Center (KAIMRC)
King Saud bin Abdulaziz University for Health Sciences
Ministry of National Guard – Health Affairs
P.O. Box 22490, Riyadh 11426
Kingdom of Saudi Arabia
Tel. No.: +966 (11) 429 4535
Fax No.: +966 (11) 429 4440
Email: matouepnasrisa@mngha.med.sa

Round 2

Reviewer 1 Report

Comments and Suggestions for Authors

The authors have not incorporated revisions addressing Questions 1, 2, and 4 raised in the previous review. For methodological reproducibility, essential experimental parameters including pre-transfection cell density, molar ratios of Cas9/sgRNA/VDR components, and detailed electroporation conditions should be explicitly provided. Additionally, the supplementary materials require revision to include an updated schematic diagram clearly depicting the CRISPR-Cas9 working mechanism.

Reviewer 4 Report

Comments and Suggestions for Authors

i am ok with the correction

Reviewer 5 Report

Comments and Suggestions for Authors

The authors took into account the comments and revised the text of the manuscript

However, reducing the size of the illustrations did not significantly improve the quality of the text on them. In the case of figures 1, 2, 3, and 6, reducing the font size on the axes and other symbols worsened their perception. The resolution of the images is insufficient. Perhaps the authors should reformat the illustrations to make them vertically oriented instead of horizontally. In its current form, the illustrative material looks bad.
